# SEMI-SUPERVISED CLIP ADAPTATION BY ENFORCING SEMANTIC AND TRAPEZOIDAL CONSISTENCY

**Kai Gan    Bo Ye    Min-Ling Zhang    Tong Wei**[*]
[1]School of Computer Science and Engineering, Southeast University, Nanjing 210096, China
[2]Key Laboratory of Computer Network and Information Integration (Southeast University),
Ministry of Education, China
{gank, yeb, zhangml, weit}@seu.edu.cn

## ABSTRACT

Vision-language pre-training models, such as CLIP, have demonstrated strong capability in rapidly adapting to downstream tasks through fine-tuning, and have been widely applied across various tasks. However, when the downstream tasks are constrained by limited image-text paired data, CLIP struggles to effectively address the domain gap between the pre-training and the target tasks. To address this limitation, we propose a novel semi-supervised CLIP adaptation method coined SEMICLIP that leverages a small amount of image-text pairs alongside a large volume of images without text descriptions to enhance CLIP's cross-modal alignment. To effectively utilize unlabeled images, we introduce semantic concept mining to improve task-specific visual representations by matching images with relevant concepts mined from labeled data. Leveraging matched semantic concepts, we construct *learnable surrogate captions* for unlabeled images and optimize a *trapezoidal consistency* to regulate the geometric structure of image-text pairs in the representation space. Experimental results demonstrate that our approach significantly improves the adaptability of CLIP in target tasks with limited labeled data, achieving gains ranging from $1.72\% - 6.58\%$ for zero-shot classification accuracy and $2.32\% - 3.23\%$ for image-text retrieval performance on standard benchmarks. The source code is available at https://github.com/Gank0078/SemiCLIP.

## 1 INTRODUCTION

Vision-language pre-training has emerged as a prominent learning paradigm, especially CLIP (Radford et al., 2021), which has benefited various downstream tasks (Li et al., 2021c; Yao et al., 2021; Gao et al., 2022; Li et al., 2021a; Gan & Wei, 2024; Xia et al., 2024) due to its advanced capabilities in image-text semantic representation. However, when adapting CLIP to specific domains such as remote sensing (Arampacha et al., 2021) and medical imaging (Kim et al., 2023) with limited image-text pairs, the model fails to effectively mitigate the domain gap, resulting in unsatisfactory performance. Previous research has primarily concentrated on the development of large-scale pre-training datasets for each domain (Johnson et al., 2019; Yang et al., 2020; Schuhmann et al., 2022), necessitating significant human and time resources. Fortunately, in most scenarios, capturing devices can readily obtain a substantial number of domain-relevant images, prompting us to consider whether leveraging a large volume of unlabeled images can enhance the generalization capabilities of vision-language pre-training for specific downstream tasks.

Semi-supervised learning (SSL) is a common way to leverage unlabeled data to improve generalization performance, and remarkable progress has been made in this research field (Sohn et al., 2020; Zhang et al., 2021; Ye et al., 2024; Chen et al., 2023). However, most existing SSL methods are designed for classification tasks and cannot be directly used in vision-language pre-training. Some research has explored the use of additional tools, such as object detectors (Li et al., 2021b; Wang et al., 2023a) and large language models (Yang et al., 2023; Mirza et al., 2024), to improve the multi-modal alignment of the CLIP model in cases where the quantity of image-text pairs is constrained. Recently, S-CLIP (Mo et al., 2023) proposes generating keyword-level and caption-level pseudo-labels for unlabeled

---

[*]Corresponding author

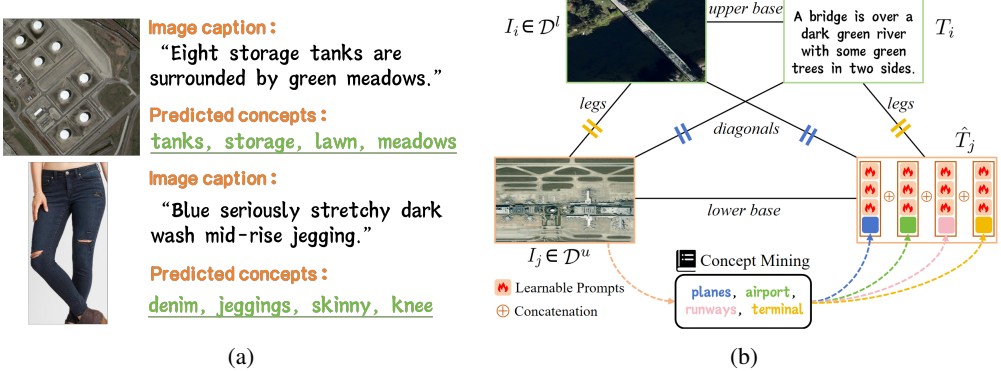

(a)                                                                    (b)

Figure 1: (1a) Relevant concepts predicted by the proposed *semantic concepts mining* module for unlabeled images. Notably, the true captions are missing in practice, and are placed in the figure merely as a reference; (1b) Illustration of our proposed *trapezoidal consistency* regularization. A learnable surrogate caption is constructed for each unlabeled image based on mined concepts.

images to enable semi-supervised CLIP adaptation. The core idea is that the true caption of unlabeled images can be linearly reconstructed from the captions of labeled data within the same mini-batch. Although S-CLIP achieves significant performance improvements compared to the original CLIP, it can only leverage the relational information within the mini-batch and is still unable to make the most effective use of all the labeled data, resulting in unsatisfactory performance in real-world scenarios.

This paper proposes a novel semi-supervised CLIP adaptation method, SEMiCLIP. For semi-supervised CLIP training, labeled and unlabeled images should share the same task domain, which suggests that a large number of common concepts can be leveraged between them. Therefore, we propose mining a candidate concept database from labeled data, and training a classifier to match images and concepts. We also generate pseudo-labels for unlabeled images using the classifier and construct learnable surrogate captions to facilitate semi-supervised training. Overall, SEMiCLIP follows the "supervised pre-training, semi-supervised fine-tuning" paradigm for SSL.

During the supervised pre-training, SEMiCLIP employs a *semantic concepts mining* module to extract candidate concepts from labeled data and learn to associate images with the concepts via a classification loss on top of the visual representations. SEMiCLIP also optimizes the image-text contrastive loss for labeled data. Once the concepts mining module is trained, we can predict relevant concepts for unlabeled images and use them as pseudo-labels to fine-tune the task-specific visual representations. As illustrated in Figure 1a, the predicted concepts demonstrate a high correlation with images, even though some concepts may not appear in the true image captions.

During the semi-supervised fine-tuning, we propose concept-level and caption-level consistency regularization to fully exploit unlabeled images. At the concept level, for unlabeled images, we optimize the model's predictions for strongly augmented images to remain consistent with those for weakly augmented images, thereby improving the model's generalization. At the caption level, we propose trapezoidal consistency to constrain the geometric relationships between image-text pairs in the representation space.

As illustrated in Figure 1b, image-text pairs in labeled data form the upper base of the trapezoid, while unlabeled images and their corresponding surrogate captions constitute the lower base. The proposed trapezoidal consistency first includes a contrastive loss which seeks to minimize the length (distance) of the upper base (representing accurate image-text pairs), and another two squared losses to restrict the trapezoid to have equal-length legs and diagonals. However, the length of the lower base (representing less accurate pairs) is adaptively adjusted by the model, which allows the model to maintain robust alignment for reliable pairs, while giving flexibility to handle noise in the less reliable pairs, particularly when dealing with unlabeled images. Our main contributions can be summarized as follows:

- We propose a novel semi-supervised CLIP adaptation method, SEMiCLIP, which mines candidate semantic concepts from labeled data and learns to associate images with concepts, providing a new way of using unlabeled images in vision-language pre-training.

- We propose the trapezoidal consistency, which uses prompts-driven templates and predicted concepts to construct surrogate captions for unlabeled images, and enhances the multi-modal alignment by exploiting the geometric structure of trapezoids in the representation space.
- Extensive experimental analyses show that SEMiCLIP achieves state-of-the-art zero-shot and image-text retrieval performance on multiple benchmarks across various domains, e.g., SEMiCLIP improves S-CLIP by over 3.75% on average in zero-shot classification tasks.

## 2 RELATED WORKS

**Vision-language pre-training** (Radford et al., 2021; Jia et al., 2021; Pham et al., 2023; Huang et al., 2023) have demonstrated remarkable success in various downstream tasks through fine-tuning. Various multiple cross-modality loss functions have been introduced as training objectives, including image-text matching (Chen et al., 2020; Li et al., 2019; Lu et al., 2019; Tan & Bansal, 2019), masked language modeling (Dong et al., 2023; Bao et al., 2022; Wang et al., 2023b), masked image modeling (Zhou et al., 2020; Xie et al., 2022), and contrasive loss (Li et al., 2021a; 2020a;b). CLIP, as one of the most representative works, aims to achieve alignment between image and text representations by learning a joint embedding. CLIP demonstrates remarkable effectiveness in tasks such as zero-shot classification (Radford et al., 2021) image-text retrieval (Luo et al., 2022), few-shot learning (Liu et al., 2022), and long-tailed learning (Shi et al., 2024).

**Semi-supervised learning** aims to improve the generalization performance by leveraging unlabeled data through entropy minimization (Grandvalet & Bengio, 2004), consistency regularization (Sohn et al., 2020), distribution alignment (Kim et al., 2020), and contrastive learning (Lee et al., 2022; Yang et al., 2022). Most of these methods focus on selecting reliable pseudo-labels throughout the training process. In particular, FixMatch (Sohn et al., 2020) selects pseudo-labels with confidence scores greater than a predetermined threshold (e.g., 0.95). However, for SSL problems in vision-language pre-training, the above methods cannot be directly applied due to the unique caption for each image. SEMiCLIP mines semantic concepts from image captions in labeled data to construct "pseudo-labels" for unlabeled images, and the pseudo-label selection and consistency regularization can be applied.

**Training CLIP with limited image-text pairs** has gained substantial attention for adapting CLIP to downstream tasks. For example, MedCLIP (Wang et al., 2022) replaces the InfoNCE loss (Oord et al., 2018) with a semantic matching loss grounded in medical knowledge to mitigate false negatives in contrastive learning. WFH (Wang et al., 2023a) generates visual hallucinations from textual inputs, which are subsequently paired with originally unpaired texts, facilitating more diverse interactions across modalities. SimVLM (Wang et al., 2021b) is trained with a prefix language modeling objective through exploiting large-scale weak supervision. However, these methods rely on additional information or models, such as class labels, pre-trained object detectors, and large language models. The method most related to our work is S-CLIP (Mo et al., 2023), which assigns pseudo-labels for unlabeled images by assuming that the true captions can be recovered by combining image captions in labeled data. Nevertheless, the performance of S-CLIP is far from satisfactory on large-scale datasets because S-CLIP finds the nearest neighbors of each unlabeled image within the current mini-batch to construct the caption, while the labeled data outside the batch is not effectively utilized.

## 3 SEMiCLIP: TOWARDS SEMI-SUPERVISED CLIP ADAPTATION

In this section, we present the SEMiCLIP approach which consists of three key modules, i.e., semantic concepts mining, concept-level semantic consistency, and caption-level trapezoidal consistency.

### 3.1 PRELIMINARIES

In general vision-language tasks (Radford et al., 2021; Jia et al., 2021), it is common to assume that there exists abundant paired images and texts $\mathcal{D}^l = \{(I_i, T_i)\}_{i=1}^N$ for contrastive learning. However, in realistic scenarios, images can be readily captured, whereas obtaining the corresponding captions entails significant costs. Usually, we can access a substantial number of unlabeled images $\mathcal{D}^u = \{I_j\}_{j=1}^M$ without captions, where $M \gg N$. The most popular vision-language model CLIP (Radford et al., 2021) learns a joint embedding space that establishes a connection between images $I_i$ and texts $T_i$. The training objective of CLIP is to maximize the cosine similarity between the embeddings of matching image-text pairs while minimizing the similarity between non-matching

pairs. Specifically, CLIP consists of a visual encoder $f_I(\cdot)$ and a text encoder $f_T(\cdot)$, so it can generate embeddings $I_i^e = f_I(I_i)$ for images and $T_i^e = f_T(T_i)$ for texts. Then the embeddings are normalized to the unit hypersphere. Specifically, the CLIP loss takes the following form:

$$\mathcal{L}_{\text{CLIP}} = -\frac{1}{2N} \sum_{i=1}^{N} \log \frac{\exp(\langle I_i^e, T_i^e \rangle / \tau)}{\sum_{t=1}^{N} \exp(\langle I_i^e, T_t^e \rangle / \tau)} + \log \frac{\exp(\langle I_i^e, T_i^e \rangle / \tau)}{\sum_{t=1}^{N} \exp(\langle I_t^e, T_i^e \rangle / \tau)} \tag{1}$$

where $\langle \cdot, \cdot \rangle$ represents the cosine similarity for normalized embeddings and $\tau$ is a temperature scaler which is simply set to 1 in our experiments. After pre-training, CLIP can perform zero-shot classification by encoding class names as text with appropriate templates (e.g., "`a photo of a {class name}`") and computing the similarities between the test image embedding and each class embedding. CLIP predicts the class associated with the highest image-text similarity.

## 3.2 SEMANTIC CONCEPTS MINING

During the supervised pre-training, SEMICLIP performs Semantic Concepts Mining (SCM), which consists of mining semantic concepts from the available captions and leveraging labeled data for CLIP adaptation. Despite the limited information contained within the texts, it is highly likely that the concepts it encompasses are also present in unlabeled images. Thus, we first employ NLTK (Bird et al., 2009) to extract the corresponding nouns from the captions in labeled data. Considering the potential noise associated with certain nouns, we filter nouns to ensure their frequency exceeds a threshold of $\mu$, which is set to 5 in our experiments. However, some nouns appear excessively often such as the word "Figure" in the SciCap dataset (Hsu et al., 2021) which contains numerous scientific figures. These overly frequent nouns (occurrence rate exceeds 30%) are not beneficial for improving performance in downstream tasks. After extraction, we recognize a total of $V$ noun entities $\{C_v\}_{v=1}^{V}$ which are regarded concepts exist within the domain corresponding to the task, and for each captioned image $I_i$, we can identify its multi-concepts encoding $y_{I_i} \in \{0,1\}^V$ where positions are set to 1 if the image contains the concept. Notably, an image can associate with multiple concepts, hence the label encoding $y_{I_i}$ can consist of more than one entry of 1s.

In semi-supervised classification tasks, models trained on labeled data can generate pseudo-labels for unlabeled data, where labels are similar to the concepts defined above. We thus inquire: *Can the model exploit concepts in captioned images to annotate corresponding concepts in unlabeled images?* For this purpose, we train a linear classifier $\psi$ with extracted concepts. Specifically, due to the rich semantic representation capabilities of the CLIP text encoder $f_T(\cdot)$, we employ the template (e.g., "`a photo includes {Concept}`") integrating the extracted concepts to generate hand-crafted descriptions and compute their features $I_{C_1}^e, \ldots, I_{C_V}^e$ which are then utilized to initialize the weights in $\psi$. Subsequently, we optimize $\psi$ through a soft cross-entropy loss to match the labeled images with their corresponding concepts as follows:

$$\mathcal{L}_{\text{SCM}} = -\sum_{i=1}^{N} \frac{y_{I_i}}{|y_{I_i}|} \log \psi(I_i^e) \tag{2}$$

where $|y_{I_i}| = \sum_{v=1}^{V} y_{I_i}(v)$ is utilized to normalize the $y_{I_i}$ so that it can conform to the properties of probabilities. Through minimizing Equation (2), $\psi$ can predict concepts for any domain-related unlabeled images. In this stage, SEMICLIP simultaneously minimizes Equation (1) and Equation (2), and in Figures 2a and 2b, we find that the alignment to concepts can facilitate the model's adaptation to the domain of training data, resulting in improved performance on downstream tasks compared to S-CLIP in some scenarios. The objective for supervised pre-training is $\mathcal{L} = \mathcal{L}_{\text{CLIP}} + \mathcal{L}_{\text{SCM}}$.

At the end of the training, we assign pseudo labels $\hat{y}_{I_j}$ which assign 1 to the positions with top-$k$ predicted confidence (Sohn et al., 2020) to unlabeled images leveraging $\psi$ to help understand the semantic information in unlabeled images and achieve further alignment in SSL fine-tuning.

## 3.3 LEARNING FROM UNLABELED IMAGES

To further improve the CLIP model for the target task, we leverage unlabeled images and semantic concepts associated with them to conduct semi-supervised fine-tuning, which includes concept-level semantic consistency and caption-level trapezoidal consistency.

### 3.3.1 CONCEPT-LEVEL SEMANTIC CONSISTENCY

When we obtain pseudo labels for unlabeled images, an intuitive strategy in SSL is to train $\psi$ for further alignment between pseudo-concepts and unlabeled images. Popular SSL methods FixMatch (Sohn et al., 2020) and ReMixMatch (Berthelot et al., 2019) align the predictions of strongly augmented images (e.g., applying RandomAugment (Cubuk et al., 2020) to images) with those of weakly augmented images, which partially mitigates confirmation bias (Arazo et al., 2020; Wang et al., 2021a) in SSL tasks and contributes to performance improvement. It is noteworthy that strong data augmentation may hurt the performance (Mu et al., 2022) for vision-language pre-training due to the alterations in the color or position of objects within the images, resulting in a mismatch between the images and their corresponding captions. However, since the concepts we consider primarily consist of nouns that denote entities, which are not significantly affected by data augmentation, making them highly suitable for concept-level semantic consistency:

$$\mathcal{L}_{\text{SCM}}^u = -\sum_{j=1}^{|\mathcal{D}^u|} \frac{\hat{y}_{I_j}}{|\hat{y}_{I_j}|} \log \psi(f_I(\Omega(I_j))) \tag{3}$$

where $\Omega(I_j)$ represents the specific strong augmentation scheme for $I_j$. Incorporating concept-level semantic consistency improves the model's robustness to spurious feature patterns (Sohn et al., 2020; Wei & Gan, 2023) and augments the model's perceptual ability regarding concepts. We will elucidate the performance benefits of employing strong data augmentation as compared to weak data augmentation in Sec. 4.6. It is noteworthy that popular strategies for semi-supervised classification, such as strong data augmentation and confidence filtering (for $\hat{y}_{I_j}$ generation), are utilized within SEMICLIP, bridging the gap between SSL classification and semi-supervised CLIP adaptation.

### 3.3.2 CAPTION-LEVEL TRAPEZOIDAL CONSISTENCY

The concepts predicted by $\psi$ are highly relevant to the unlabeled images, revealing the information of entities contained in the images. However, relying solely on the concepts present in the image is insufficient to effectively summarize all the content contained within the image due to the absence of descriptive attributes for the concepts and their interrelations. Therefore, how to aggregate concepts appropriately constitutes a crucial challenge.

Inspired by the remarkable performance of context optimization (Zhou et al., 2022b;a) for automated prompt engineering, we introduce learnable prompts which can integrate and link corresponding concepts in the images, striving to create meaningful connections between the various concepts. Specifically, we leverage the prompts-driven template $[V]_1[V]_2[V]_3[Concept]$ to describe each concept in the image, where $[V]_{1:3}$ represent learnable vectors that establish connections among various concepts through optimization during training. To expedite the model's training adaptation, we employ the token embeddings of "a photo includes" to initialize $[V]_{1:3}$. Considering the presence of multiple concepts in each image, we concatenate their prompts to construct a surrogate caption:

$$\hat{T}_j = [V]_1^{(1)}[V]_2^{(1)}[V]_3^{(1)}[C_{s_1}]\cdots[V]_1^{(k)}[V]_2^{(k)}[V]_3^{(k)}[C_{s_k}]. \tag{4}$$

We select the top-$k$ concepts ($C_{s_{1:k}}$) extracted from $I_j$ with the highest classification confidence (Sohn et al., 2020) in $\psi(I_j^e)$, where $s_{1:k}$ represent the indexes for top-$k$ concepts.

Although learnable prompts allow us to generate more accurate surrogate captions for unlabeled images, these captions can be coarse-grained. Hence, we adopt caption-level trapezoidal consistency, which includes constraints for the upper base, legs, and diagonals of the trapezoid constructed through the labeled image-text pairs, unlabeled images, and surrogate captions. Specifically, we select a subset of unlabeled images $I^P$ with top-$P\%$ cosine similarity between images and their corresponding surrogate captions, denoted as $\mathcal{D}^P = \{I_j^P, \hat{T}_j\}_{j=1}^{M_P}$, where $M_P = M * P\%$ represents the number of selected unlabeled images. Subsequently, we define $\widetilde{\mathcal{D}} = \mathcal{D}^l \cup \mathcal{D}^P = \{(\widetilde{I}_i, \widetilde{T}_i)\}_{i=1}^{N+M_P}$ and the trapezoidal consistency as follows:

$$\mathcal{L}_{\text{Trap}} = \underbrace{\mathcal{L}_{\text{CLIP}}}_{upper\ base} + \frac{1}{|\widetilde{\mathcal{D}}|}\sum_{i=1}^{|\widetilde{\mathcal{D}}|}\sum_{j=1}^{|\widetilde{\mathcal{D}}|} \underbrace{(\langle \tilde{I}_i^e, \tilde{T}_j^e \rangle - \langle \tilde{I}_j^e, \tilde{T}_i^e \rangle)^2}_{diagonals} + \underbrace{(\langle \tilde{I}_i^e, \tilde{I}_j^e \rangle - \langle \tilde{T}_j^e, \tilde{T}_i^e \rangle)^2}_{legs} \tag{5}$$

The diagonal constraint ensures consistent similarity across multiple image-text pairs. Simultaneously, the equal length of legs constraint ensures that the similarity between images and the similarity between texts are treated equally in the feature space.

To understand why trapezoidal consistency works for surrogate captions, we draw inspiration from two aspects:

- As illustrated in Figure 1b, the consistency for diagonals constrains the distance between $I_i$ and $\hat{T}_j$ to be consistent with the distance between $I_j$ and $T_i$. Similarly, The consistency for legs regulates the distance between $I_i$ and $I_j$ to align with the distance between $T_i$ and $\hat{T}_j$. Equation (1) encourages $I_i$ and $T_i$ to be closer, leading to a corresponding decrease in the length of the upper base. The quadrilateral characterized by vertices $I_i$, $I_j$, $T_i$, and $\hat{T}_j$ will be optimized toward an isosceles trapezoid, which does not need any distance constraints regarding the length of its lower base. This is equivalent to imposing no direct alignment constraint between $I_j$ and $\hat{T}_j$, thereby mitigating the negative impact of inaccurate $\hat{T}_j$.
- Despite the absence of direct alignment between $I_j$ and $\hat{T}_j$, the trapezoid consistency allows both to interact with samples from within and outside the modality (Goel et al., 2022), ensuring coherence among samples within each modality and substantially enhancing the model's overall alignment ability.

Overall, during semi-supervised fine-tuning, concept-level semantic consistency and caption-level trapezoidal consistency are jointly optimized by minimizing: $\mathcal{L} = \mathcal{L}_{\text{SCM}}^u + \mathcal{L}_{\text{Trap}}$.

# 4 EXPERIMENTS

## 4.1 EXPERIMENT SETUP

**Datasets and Evaluation Metrics.** We conduct extensive experiments on four publicly available datasets to evaluate the performance of SEMICLIP. Following previous method S-CLIP (Mo et al., 2023), the datasets include Remote sensing datasets (Yang & Newsam, 2010; Zhang et al., 2014; Lu et al., 2017), Fashion datasets (Han et al., 2017; Rostamzadeh et al., 2018; Vasileva et al., 2018), SciCap dataset (Hsu et al., 2021), and Simpsons dataset (Attia, 2018; Adler, 2023). Under the default setting, we subsample 10% image-text pairs of the training dataset randomly as labeled data, leaving the rest as unlabeled data. The models are evaluated on zero-shot classification and image-text retrieval tasks, with performance measured by Top-1 classification accuracy (%) and recall at $k$ (R@$k$). Results are reported as the mean and standard deviation across three random seeds.

**Implementation details.** We use the publicly available CLIP model (Ilharco et al., 2021) as our backbone. All experiments are conducted on four NVIDIA A6000 GPUs with a batch size of 64 per GPU. To maintain equitable memory usage in SSL, each mini-batch consists of 32 image-text pairs and 32 unpaired images. We utilize ViT-B-16 as the default vision encoder in our experiments, and experiments related to other vision encoders are shown in Appendix A.1. We train the model 25 epochs in the supervised pre-training, and 15 epochs for semi-supervised fine-tuning. We employ AdamW (Loshchilov, 2017) alongside a weight decay set at $5 \times 10^{-4}$ and apply the default cosine learning rate scheduling with warmup for the first 10 steps. The learning rate is set to $5 \times 10^{-5}$ for remote sensing and fashion datasets and $1 \times 10^{-6}$ for SciCap and Simpsons datasets. We establish default values of 30 for $P$ and 4 for $k$.

**Competing Methods.** We compare our method with the pre-trained CLIP, denoted as CLIP (original), and the CLIP model fine-tuned using only image-text pairs in labeled data, denoted as CLIP (fine-tuned). Oracle (fully supervised fine-tuned) is the supervised learning on all training data. We also compare SEMICLIP with another three semi-supervised CLIP adaptation approaches, i.e., Hard-PL, Soft-PL, and S-CLIP. More details about these methods can be found in Appendix B.

## 4.2 REMOTE SENSING DATASETS

Following the settings in Arampacha et al. (2021), we train vision-language models on a merged dataset, termed RS-ALL, which consists of RSICD (Lu et al., 2017), UCM (Yang & Newsam, 2010), and Sydney (Zhang et al., 2014). Considering the potentially broader domain distribution of unlabeled data, we not only utilize the default 90% of RS-ALL designated as unlabeled data (L=U), but also incorporate the RESISC45 dataset (Cheng et al., 2017) as unlabeled data (L≠U).

**Zero-shot classification.** We utilize the validation sets from the classification variants of the RSICD and UCM datasets, referred to as RSICD-CLS and UCM-CLS, respectively. Furthermore, we

| Method | Data | RSICD-CLS | UCM-CLS | WHU-RS19 | RSSCN7 | AID |
|---|---|---|---|---|---|---|
| CLIP (original) | - | 59.2 | 60.2 | 81.2 | 69.0 | 59.6 |
| CLIP (fine-tuned) | L | $75.7_{\pm2.2}$ | $80.1_{\pm6.8}$ | $92.2_{\pm1.5}$ | $70.7_{\pm5.5}$ | $79.8_{\pm3.8}$ |
| Oracle (fully supervised fine-tuned) | L + U | $87.5_{\pm1.8}$ | $67.9_{\pm2.3}$ | $94.2_{\pm1.5}$ | $75.7_{\pm2.9}$ | $89.3_{\pm1.2}$ |
| Hard-PL (Lee et al., 2013) | | $78.0_{\pm2.0}$ (+2.3) | $72.4_{\pm5.2}$ (-7.7) | $92.3_{\pm2.6}$ (+0.1) | $71.7_{\pm4.1}$ (+1.0) | $81.4_{\pm2.2}$ (+1.6) |
| Soft-PL (Assran et al., 2021) | L=U | $81.7_{\pm0.6}$ (+6.0) | $78.9_{\pm5.2}$ (-1.2) | $95.3_{\pm0.8}$ (+3.1) | $72.0_{\pm2.5}$ (+1.3) | $85.8_{\pm1.6}$ (+6.0) |
| S-CLIP (Mo et al., 2023) | | $81.4_{\pm1.8}$ (+5.7) | $81.3_{\pm3.4}$ (+1.2) | $95.9_{\pm1.8}$ (+3.7) | $75.1_{\pm2.0}$ (+4.4) | $86.4_{\pm1.7}$ (+6.6) |
| SEMiCLIP (ours) | | $\mathbf{83.8}_{\pm1.0}$ (+8.1) | $\mathbf{85.6}_{\pm3.4}$ (+5.5) | $\mathbf{96.6}_{\pm1.2}$ (+4.4) | $\mathbf{75.3}_{\pm1.9}$ (+4.6) | $\mathbf{87.4}_{\pm0.5}$ (+7.6) |
| Hard-PL (Lee et al., 2013) | | $79.9_{\pm2.5}$ (+4.2) | $76.2_{\pm1.9}$ (-3.9) | $92.3_{\pm2.6}$ (+0.1) | $71.4_{\pm2.2}$ (+0.7) | $82.4_{\pm2.0}$ (+2.6) |
| Soft-PL (Assran et al., 2021) | L≠U | $78.3_{\pm3.8}$ (+2.6) | $78.3_{\pm3.4}$ (-1.8) | $95.3_{\pm1.3}$ (+3.1) | $73.8_{\pm2.7}$ (+3.1) | $82.7_{\pm3.7}$ (+2.9) |
| S-CLIP (Mo et al., 2023) | | $78.3_{\pm3.2}$ (+2.6) | $79.5_{\pm4.9}$ (-0.6) | $93.8_{\pm1.5}$ (+1.6) | $73.9_{\pm5.0}$ (+3.2) | $84.9_{\pm3.3}$ (+5.1) |
| SEMiCLIP (ours) | | $\mathbf{84.1}_{\pm0.5}$ (+8.4) | $\mathbf{85.4}_{\pm2.8}$ (+5.3) | $\mathbf{96.5}_{\pm1.7}$ (+4.3) | $72.9_{\pm3.5}$ (+2.2) | $\mathbf{86.2}_{\pm1.0}$ (+6.4) |

Table 1: Zero-shot classification results on remote sensing datasets. We compare the original CLIP, supervised CLIP fine-tuned on labeled data (L), and semi-supervised methods that utilize unlabeled data sampled from the same (L=U) or different (L≠U) distribution as the labeled data. Parentheses indicate the performance gap from the supervised CLIP, where values highlighted in green indicate gaps larger than one. The best results are in **bold**.

| Method | Data | Image→text R@5 | | | Text→image R@5 | | |
|---|---|---|---|---|---|---|---|
| | | RSICD | UCM | Sydney | RSICD | UCM | Sydney |
| CLIP (original) | - | 12.6 | 46.7 | 44.8 | 13.9 | 39.5 | 44.8 |
| CLIP (fine-tuned) | L | $25.7_{\pm2.3}$ | $55.2_{\pm0.8}$ | $46.6_{\pm3.0}$ | $24.9_{\pm0.6}$ | $56.3_{\pm1.6}$ | $49.4_{\pm2.0}$ |
| CLIP (fine-tuned) | L + U | $34.3_{\pm1.3}$ | $63.5_{\pm1.3}$ | $63.6_{\pm2.4}$ | $32.0_{\pm1.3}$ | $66.6_{\pm1.1}$ | $65.4_{\pm2.5}$ |
| Hard-PL (Lee et al., 2013) | | $27.3_{\pm1.1}$ | $54.8_{\pm1.6}$ | $52.3_{\pm3.6}$ | $24.2_{\pm1.3}$ | $54.6_{\pm1.0}$ | $55.2_{\pm6.9}$ |
| Soft-PL (Assran et al., 2021) | L=U | $27.3_{\pm0.9}$ | $54.8_{\pm2.2}$ | $52.3_{\pm2.0}$ | $26.1_{\pm1.7}$ | $55.6_{\pm4.0}$ | $56.3_{\pm5.3}$ |
| S-CLIP (Mo et al., 2023) | | $27.5_{\pm1.1}$ | $57.0_{\pm1.5}$ | $51.1_{\pm4.3}$ | $25.6_{\pm0.6}$ | $57.3_{\pm4.1}$ | $50.6_{\pm2.6}$ |
| SEMiCLIP (ours) | | $\mathbf{27.6}_{\pm1.6}$ | $\mathbf{59.2}_{\pm2.2}$ | $\mathbf{53.4}_{\pm4.6}$ | $\mathbf{26.4}_{\pm0.4}$ | $\mathbf{58.4}_{\pm2.4}$ | $\mathbf{58.0}_{\pm2.0}$ |
| Hard-PL (Lee et al., 2013) | | $25.9_{\pm1.6}$ | $52.7_{\pm2.2}$ | $50.6_{\pm7.0}$ | $22.4_{\pm1.5}$ | $54.3_{\pm1.4}$ | $51.7_{\pm5.2}$ |
| Soft-PL (Assran et al., 2021) | L≠U | $27.4_{\pm1.6}$ | $54.1_{\pm2.4}$ | $51.1_{\pm1.0}$ | $24.4_{\pm0.7}$ | $54.4_{\pm2.4}$ | $52.3_{\pm5.3}$ |
| S-CLIP (Mo et al., 2023) | | $27.2_{\pm0.6}$ | $55.7_{\pm3.0}$ | $50.0_{\pm4.6}$ | $26.1_{\pm0.7}$ | $55.7_{\pm2.9}$ | $52.9_{\pm5.5}$ |
| SEMiCLIP (ours) | | $\mathbf{28.5}_{\pm0.2}$ | $\mathbf{59.5}_{\pm3.6}$ | $\mathbf{54.6}_{\pm3.6}$ | $\mathbf{26.3}_{\pm0.4}$ | $\mathbf{59.5}_{\pm1.3}$ | $\mathbf{58.6}_{\pm4.6}$ |

Table 2: Image-text retrieval results on remote sensing datasets, following the same setup of Table 1.

assess the generalization capability of SEMiCLIP on WHU-RS19 (Xia et al., 2009), RSSCN7 (Zou et al., 2015), and AID (Xia et al., 2017), which are not included in the training datasets. Results in Tab. 1 demonstrate the performance advantages of SEMiCLIP. Compared with S-CLIP, SEMiCLIP demonstrates an average performance gain of 1.72% and 2.94% for L=U and L≠U settings, respectively. L≠U represents the scenarios that demand higher robustness of the model against unseen domains and domain shifts, where SEMiCLIP achieves more significant performance gains compared to previous methods. Interestingly, our method even outperforms the oracle in some zero-shot settings, and we provide more detailed explanation of this phenomenon in Appendix E.

**Image-text retrieval.** We evaluate the image→text and text→image retrieval performance on the validation sets of the RSICD, UCM, and Sydney datasets. Tab. 2 presents the R@5 results, where we can observe that the proposed SEMiCLIP surpasses all competing methods across all settings, achieving average 4.48% gain over fine-tuned CLIP. More results can be found in Appendix A.3.

### 4.3 FASHION DATASETS

We evaluate the models on the union of Fashion200k (Han et al., 2017), FashionGen (Rostamzadeh et al., 2018), and Polyvore Outfits (Vasileva et al., 2018) datasets. Given that each text caption can associate with multiple image views, we randomly select one image during each iteration.

**Zero-shot classification.** The zero-shot classification performance is evaluated on the classification validation sets of Fashion200k, FashionGen, and Polyvore Outfits. It is worth noting that we considered two different class hierarchies, namely super-class and sub-class, in the Fashion200k and FashionGen datasets. The results in Tab. 3 indicate that SEMiCLIP significantly outperforms S-CLIP, with an average gain of 12.15% on sub-class in particular. We attribute this to the richness of the concepts extracted by SEMiCLIP through semantic concepts mining, and the concepts may encom-

| Method | Fashion200k | | FashionGen | | Polyvore |
| --- | --- | --- | --- | --- | --- |
| | Super-class | Sub-class | Super-class | Sub-class | Class |
| CLIP (original) | 73.2 | 27.7 | 34.8 | 26.4 | 70.2 |
| CLIP (fine-tuned) | 79.0±3.5 | 35.1±0.7 | 35.4±8.1 | 24.5±2.4 | 60.4±2.3 |
| Oracle (fully supervised fine-tuned) | 88.6±1.3 | 45.3±4.8 | 53.3±2.5 | 45.5±7.3 | 76.3±8.8 |
| Hard-PL (Lee et al., 2013) | 54.9±6.4 (-24.1) | 23.9±2.3 (-11.2) | 24.2±3.9 (-11.2) | 18.3±2.4 (-6.2) | 34.3±6.8 (-26.1) |
| Soft-PL (Assran et al., 2021) | 82.5±2.8 (+3.5) | 36.6±1.4 (+1.5) | 44.8±3.5 (+9.4) | 33.6±1.4 (+9.1) | 73.6±1.7 (+13.2) |
| S-CLIP (Mo et al., 2023) | 85.1±0.9 (+6.1) | 38.4±0.7 (+3.3) | 44.0±4.6 (+8.6) | 29.6±4.0 (+5.1) | 73.9±2.4 (+13.5) |
| SEMICLIP (ours) | **85.8**±0.4 (+6.8) | **44.7**±1.0 (+9.6) | **51.4**±0.6 (+16.0) | **47.6**±1.7 (+23.1) | **74.4**±0.5 (+14.0) |

Table 3: Zero-shot classification results on fashion datasets. Parentheses indicate the performance gap from the supervised CLIP, where values highlighted in green indicate gaps larger than one.

| Method | Image→text R@5 | | | Text→image R@5 | | |
| --- | --- | --- | --- | --- | --- | --- |
| | Fashion200k | FashionGen | Polyvore | Fashion200k | FashionGen | Polyvore |
| CLIP (original) | 12.5 | 23.5 | 24.3 | 12.3 | 27.7 | 27.4 |
| CLIP (fine-tuned) | 13.7±0.4 | 32.1±0.2 | 16.3±0.5 | 13.5±0.2 | 31.9±0.2 | 16.2±0.3 |
| Oracle (fully supervised fine-tuned) | 34.9±0.7 | 63.4±0.1 | 48.5±0.8 | 34.9±0.9 | 35.3±0.9 | 49.3±0.5 |
| Hard-PL (Lee et al., 2013) | 12.6±0.7 | 26.8±0.7 | 14.9±0.8 | 14.5±0.8 | 30.2±0.4 | 17.3±0.9 |
| Soft-PL (Assran et al., 2021) | 15.2±0.8 | 35.0±0.3 | 18.8±0.3 | 15.9±0.7 | 37.4±0.7 | 21.9±0.2 |
| S-CLIP (Mo et al., 2023) | 16.9±0.4 | 37.9±0.4 | 22.6±0.4 | 17.4±0.3 | 40.9±0.4 | 24.2±0.5 |
| SEMICLIP (ours) | **22.1**±0.4 | **48.4**±0.7 | **27.7**±0.3 | **22.5**±0.2 | **49.7**±0.3 | **27.6**±0.2 |

Table 4: Image-text retrieval results on fashion datasets, following the same setup of Table 3.

pass some fine-grained information. Coupled with subsequent concepts-level semantic consistency, SEMICLIP can significantly improve fine-grained classification performance.

**Image-text retrieval.** We evaluate the image-text retrieval performance on the validation sets of three fashion datasets. We can observe that SEMICLIP outperforms S-CLIP by an average gain of 6.35% in terms of R@5 in Tab. 4. Compared to results for remote sensing datasets, it is evident that SEMICLIP exhibits a more pronounced advantage, as the Fashion datasets present greater challenges with large-scale and diverse samples. Such settings are more representative of real-world scenarios. Methods like S-CLIP, which utilize nearest neighbor strategies for pseudo-labeling, often struggle to find suitable neighbors within a single batch.

## 4.4 MORE DATASETS

We conduct experiments on SciCap (Hsu et al., 2021) and Simpsons (Adler, 2023) datasets to further verify the performance superiority in image-text retrieval.

**SciCap.** The images in SciCap belong to the scientific figures domain, which presents a significant challenge for the model. The results in Tab. 5 indicate that SEMICLIP improve the performance averaged 4.03% compared to fine-tuned CLIP. This further confirms that SEMICLIP can enhance the performance of pre-trained models by leveraging a substantial amount of unlabeled data, even in challenging domain settings.

**Simpsons.** Due to the Simpsons dataset only contains limited image-text pairs, we integrate unlabeled images from an additional dataset called Simpsons Characters (Attia, 2018). From Tab. 5, we can conclude that SEMICLIP achieves averaged 6.55% and 6.95% enhancement for R@1 and R@5 respectively compared to S-CLIP. The results demonstrate that even when image-text pairs are extremely scarce, SEMICLIP can still maintain a significant performance advantage.

## 4.5 IN-DEPTH ANALYSIS FOR SEMICLIP

In this section, we conduct extensive studies to demonstrate the effectiveness of SEMICLIP and analysis for the sensitivity of hyperparameters can be found in Appendix A.2.

**Extracted concepts.** In Figure 1a, we provide examples illustrating the concepts extracted by the semantic concepts extractor for the unlabeled images. We note that some extracted concepts, like "tanks" and "jeggings", are indeed present in the corresponding real captions of the images. However, there are also concepts like "lawn" and "skinny" that do not appear in the captions, but these non-

| | SciCap | | | | Simpsons | | | |
|---|---|---|---|---|---|---|---|---|
| | Image→text | | Text→image | | Image→text | | Text→image | |
| Method | R@1 | R@5 | R@1 | R@5 | R@1 | R@5 | R@1 | R@5 |
| CLIP (original) | 8.3 | 13.8 | 8.8 | 14.2 | 13.2 | 35.5 | 10.5 | 32.9 |
| CLIP (fine-tuned) | $11.0_{\pm0.6}$ | $19.0_{\pm0.8}$ | $11.1_{\pm0.9}$ | $18.1_{\pm0.0}$ | $19.3_{\pm2.0}$ | $48.2_{\pm6.0}$ | $14.5_{\pm0.0}$ | $45.6_{\pm2.0}$ |
| Oracle (fully supervised fine-tuned) | $14.8_{\pm0.9}$ | $25.5_{\pm1.4}$ | $14.8_{\pm0.5}$ | $25.7_{\pm0.3}$ | $28.3_{\pm3.7}$ | $52.3_{\pm5.6}$ | $22.3_{\pm3.1}$ | $55.8_{\pm4.1}$ |
| Hard-PL (Lee et al., 2013) | $11.2_{\pm0.2}$ | $18.9_{\pm0.2}$ | $12.2_{\pm0.1}$ | $20.1_{\pm0.1}$ | $16.7_{\pm2.7}$ | $42.1_{\pm5.3}$ | $15.4_{\pm4.2}$ | $43.9_{\pm3.3}$ |
| Soft-PL (Assran et al., 2021) | $11.5_{\pm0.2}$ | $19.4_{\pm0.3}$ | $12.2_{\pm0.2}$ | $20.5_{\pm0.2}$ | $18.4_{\pm1.3}$ | $40.4_{\pm4.0}$ | $15.8_{\pm1.3}$ | $43.4_{\pm3.5}$ |
| S-CLIP (Mo et al., 2023) | $11.6_{\pm0.5}$ | $20.6_{\pm0.5}$ | $12.6_{\pm0.4}$ | $21.1_{\pm0.4}$ | $15.4_{\pm2.0}$ | $43.9_{\pm5.5}$ | $17.5_{\pm1.5}$ | $47.4_{\pm6.8}$ |
| SEMICLIP (ours) | $\mathbf{12.3}_{\pm0.4}$ | $\mathbf{21.0}_{\pm0.4}$ | $\mathbf{13.0}_{\pm0.0}$ | $\mathbf{21.5}_{\pm0.2}$ | $\mathbf{24.1}_{\pm0.8}$ | $\mathbf{50.4}_{\pm2.7}$ | $\mathbf{21.9}_{\pm1.5}$ | $\mathbf{54.8}_{\pm0.8}$ |

Table 5: Image-text retrieval results on SciCap (scientific figures) and Simpsons (comics) datasets. We train the model on each dataset and evaluate on their validation sets.

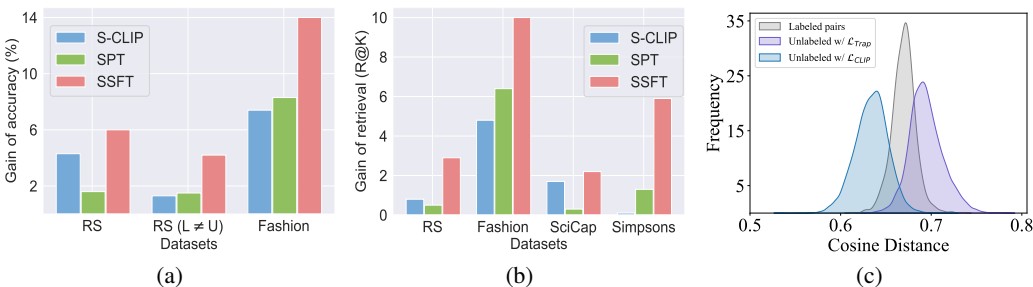

(a)             (b)             (c)

Figure 2: (2a) Gain of averaged zero-shot classification for S-CLIP and different stages in SEMICLIP. "SPT" and "SSFT" respectively denote supervised pre-training and semi-supervised fine-tuning; (2b) Gain of averaged image-text retrieval for S-CLIP and different stages in SEMICLIP; (2c) The cosine distance for image-text pairs. "Labeled pairs" and "Unlabeled w/ $\mathcal{L}_{\text{Trap}}$" denote the distance between images and captions in labeled data and the distance between unlabeled images and surrogate captions when training with $\mathcal{L}_{\text{Trap}}$. "Unlabeled w/ $\mathcal{L}_{\text{CLIP}}$" represent the distance when directly aligning unlabeled images and surrogate captions through CLIP loss.

existent concepts closely align with the content of the images. This indicates that the classifier $\psi$ can effectively perceive the concepts in images, which is beneficial for generating surrogate captions.

**Gain of performance.** Figures 2a and 2b illustrate the performance gains of S-CLIP and SEMICLIP at different stages compared to fine-tuned CLIP. It is evident that all three methods surpass fine-tuned CLIP to some extent, and SEMICLIP in semi-supervised fine-tuning shows a significant performance advantage over S-CLIP. Surprisingly, in SPT, SEMICLIP can completely outperform fine-tuned CLIP and even surpass S-CLIP in some settings, such as zero-shot and image-text retrieval performance in fashion datasets. This indicates that leveraging concepts extracted from captions and trained by Equation (2) can enhance the model's perceptual capability regarding concepts which lays a solid foundation for the subsequent semi-supervised fine-tuning.

**Structure of trapezoid.** As described in Sec. 3.3.2, the Equation (5) will optimize the structure illustrated in Figure 1b to resemble an isosceles trapezoid. In Figure 2c, we present the frequency distribution of the cosine distances between labeled image-text pairs and the cosine distances between unlabeled images and surrogate captions. We observe that the distance between unlabeled images and surrogate captions is generally larger when we apply $\mathcal{L}_{\text{Trap}}$, indicating that the lower base is longer than the upper base. However, if CLIP loss is directly imposed on unlabeled images and surrogate captions, it causes their distances to become excessively close, resulting in a significant decline in zero-shot classification by 4.07% on average. This indicates that the direct alignment of unlabeled images and surrogate captions suffers from performance degradation due to the noise present in surrogate captions, whereas trapezoidal consistency effectively mitigates this issue and significantly improves performance through interactions between samples from both image and text modalities.

### 4.6 ABLATION ANALYSIS

We tease apart the factors that contribute significantly to the success of SEMICLIP in Tab. 6.

| Ablations | Remote sensing | | | Fashion | | |
|---|---|---|---|---|---|---|
| | ZS | I2T | T2I | ZS | I2T | T2I |
| SEMICLIP | $85.7_{\pm1.8}$ | $32.4_{\pm1.8}$ | $31.2_{\pm0.9}$ | $60.8_{\pm1.3}$ | $24.1_{\pm1.1}$ | $24.5_{\pm1.5}$ |
| w/o Strong augmentation | $84.4_{\pm1.9}$ | $31.9_{\pm1.9}$ | $30.3_{\pm1.5}$ | $57.7_{\pm2.2}$ | $22.3_{\pm1.9}$ | $22.5_{\pm1.7}$ |
| w/o CSC | $82.7_{\pm1.8}$ | $32.0_{\pm2.1}$ | $30.6_{\pm1.4}$ | $59.1_{\pm1.4}$ | $23.5_{\pm2.1}$ | $23.9_{\pm1.2}$ |
| w/o Consistency for diagonals | $82.0_{\pm2.6}$ | $31.9_{\pm0.9}$ | $30.8_{\pm1.1}$ | $56.4_{\pm1.5}$ | $22.6_{\pm1.3}$ | $23.9_{\pm1.8}$ |
| w/o Consistency for legs | $83.8_{\pm2.4}$ | $31.4_{\pm3.7}$ | $30.5_{\pm2.1}$ | $59.7_{\pm1.2}$ | $23.8_{\pm1.1}$ | $24.0_{\pm2.1}$ |
| w/o CTC | $81.9_{\pm1.8}$ | $31.7_{\pm2.8}$ | $30.7_{\pm1.6}$ | $54.1_{\pm1.4}$ | $22.1_{\pm2.3}$ | $22.4_{\pm1.5}$ |
| w/o Learnable prompts | $84.6_{\pm2.4}$ | $31.5_{\pm1.2}$ | $30.5_{\pm1.7}$ | $59.3_{\pm1.9}$ | $23.2_{\pm1.7}$ | $23.8_{\pm2.3}$ |

Table 6: Ablation studies. We investigate the impact of the core components of the SEMICLIP on remote sensing and fashion datasets. "ZS", "I2T", "T2I" denote the averaged zero-shot classification, image→text retrieval and text→image retrieval performance. "CSC" and "CTC" represent the concept-level semantic consistency and caption-level trapezoidal consistency, respectively.

**Impact of strong data augmentation.** We apply strong data augmentation to unlabeled images during the concept-level semantic consistency in Equation (3). Upon replacing strong augmentation with weak augmentation, we notice a significant performance decline across all settings, i.e., 3.1% zero-shot accuracy drop in fashion datasets. It suggests that strong data augmentation significantly contributes to the improvement of zero-shot performance.

**Impact of concept-level semantic consistency (CSC).** CSC is employed to further leverage extracted concepts to enhance the model's awareness of concepts in task-specific domains. The exclusion of CSC results in averaged 1.15% performance drop. It is noteworthy that in some cases, the removal of strong augmentation leads to a more severe performance decline than the removal of CSC (e.g., the "ZS" in fashion). We believe this is due to the confirmation bias when applying weak augmentation for consistency, which can cause the model's biases to accumulate, severely impairing performance.

**Impact of consistency for the diagonals of trapezoid.** The consistency for diagonals can symmetrize the similarity across two mismatched image-text pairs. When removing the consistency for diagonals, we observe a reduction in performance averaging 1.85%. This suggests that cross-modal interaction and consistency constraints are beneficial for enhancing the model's alignment capabilities.

**Impact of consistency for the legs of trapezoid.** SEMICLIP utilizes consistency for legs to enhance the intra-modality similarity. While the benefits derived from the consistency for legs are relatively small compared to the diagonals, with an average performance decrease of 0.92% upon its removal, it nevertheless underscores the potential of constraints for legs to enhance overall performance by ensuring coherence among samples within each modality.

**Impact of caption-level trapezoidal consistency (CTC).** The complete removal of CTC undoubtedly leads to a significant performance decline, with reductions of 5.25% and 1.33% observed on the zero-shot and image-text retrieval tasks, respectively. This highlights that CTC plays an important role in strengthening the interaction and alignment between samples. More importantly, it facilitates the integration of surrogate captions into the training, further leveraging unlabeled images to improve the performance and effectively mitigating noise potentially introduced by surrogate captions.

**Impact of learnable prompts in surrogate caption.** In order to investigate the role of learnable prompts in constructing surrogate captions for unlabeled images, we detached the gradients and froze the prompts after initialization. Results reveal that applying learnable prompts can achieve an averaged 0.97% improvement in model performance, which we believe stems from the dynamic adaptation of prompts that better aligns surrogate captions with task-specific domains.

## 5 CONCLUSION

This paper proposes a new semi-supervised CLIP adaptation method that effectively adapts CLIP to target tasks using only a small number of image-text pairs. The proposed trapezoidal consistency can effectively tackle inaccurate surrogate captions, achieving robust representation learning. Extensive experiments demonstrate that our method achieves state-of-the-art results in both zero-shot classification and image-text retrieval tasks. We believe that trapezoidal consistency has the potential to flexibly adapt to different multi-modal semi-supervised learning tasks.

ACKNOWLEDGMENTS

This work was supported by the National Science Foundation of China (62206049, 62176055), Postgraduate Research&Practice Innovation Program of Jiangsu Province (KYCX25_0544), and the Big Data Computing Center of Southeast University. We would like to thank anonymous reviewers for their constructive suggestions.

**Ethics Statement.** This research fully complies with the specified ethical standards. No human participants or sensitive data were used, and the datasets utilized are open and ethically sourced. The research methodology and results have been carefully evaluated to prevent any potential misuse, discrimination, or bias. All procedures align with legal, privacy, and security regulations. There are no conflicts of interest or external funding that could influence the study's findings. We have maintained transparency and integrity throughout the research process to ensure that the results are accurately and ethically represented.

**Reproducibility Statement.** We have made significant efforts to ensure the reproducibility of our results. The source code used in our experiments is included in the supplementary materials, along with a detailed README file that provides step-by-step instructions and the necessary commands to reproduce the experiments. All the hyperparameters and experimental settings are specified in Sec. 4.1 to facilitate replication.

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

# A    ADDITIONAL EXPERIMENTS

## A.1    DIFFERENT NEURAL NETWORK ARCHITECTURES

We conduct experiments to evaluate the performance of SEMICLIP in zero-shot classification and image-text retrieval with different neural network architectures. As shown in Tabs. 7 and 8, SEMICLIP achieves an average performance improvement of 3.52% on zero-shot classification and 0.28% on image-text retrieval compared to S-CLIP when adopting ResNet-50 as the backbone. In addition, SEMICLIP exhibits averaged performance advantage of 2.62%, 4.20%, and 1.10% over S-CLIP in zero-shot classification, image→text retrieval, and text→image retrieval when implemented with the ViT-B-32 architecture. This indicates that SEMICLIP can achieve consistent and significant performance advantages for different neural network architectures and model sizes.

| Model | Method | RSICD-CLS | UCM-CLS | WHU-RS19 | RSSCN7 | AID |
|---|---|---|---|---|---|---|
| ResNet-50 | CLIP (original) | 45.3 | 50.5 | 65.5 | 58.9 | 47.8 |
| | CLIP (fine-tuned) | $58.3_{\pm 0.3}$ | $63.5_{\pm 3.4}$ | $76.5_{\pm 3.2}$ | $61.9_{\pm 1.2}$ | $63.1_{\pm 1.3}$ |
| | S-CLIP | $66.9_{\pm 1.7}$ (+8.6) | $66.7_{\pm 1.6}$ (+3.2) | $86.9_{\pm 2.0}$ (+10.4) | $66.2_{\pm 1.1}$ (+4.3) | $73.0_{\pm 0.3}$ (+9.9) |
| | SEMICLIP | $\mathbf{72.0}_{\pm 2.1}$ (+13.7) | $\mathbf{73.6}_{\pm 1.4}$ (+10.1) | $\mathbf{87.2}_{\pm 2.4}$ (+10.7) | $\mathbf{68.9}_{\pm 0.4}$ (+7.0) | $\mathbf{75.6}_{\pm 1.7}$ (+12.5) |
| ViT-B-32 | CLIP (original) | 56.2 | 58.5 | 76.3 | 62.3 | 55.6 |
| | CLIP (fine-tuned) | $73.6_{\pm 3.1}$ | $79.0_{\pm 4.4}$ | $91.1_{\pm 1.3}$ | $70.4_{\pm 3.1}$ | $79.6_{\pm 2.5}$ |
| | S-CLIP | $78.5_{\pm 0.5}$ (+4.9) | $77.4_{\pm 4.4}$ (-1.6) | $94.9_{\pm 0.5}$ (+3.8) | $71.1_{\pm 2.6}$ (+0.7) | $84.2_{\pm 1.6}$ (+4.6) |
| | SEMICLIP | $\mathbf{81.3}_{\pm 0.5}$ (+7.7) | $\mathbf{84.3}_{\pm 3.9}$ (+5.3) | $\mathbf{95.2}_{\pm 0.8}$ (+4.1) | $\mathbf{73.0}_{\pm 0.6}$ (+2.6) | $\mathbf{85.4}_{\pm 0.3}$ (+5.8) |
| ViT-B-16 | CLIP (original) | 59.2 | 60.2 | 81.2 | 69.0 | 59.6 |
| | CLIP (fine-tuned) | $75.7_{\pm 2.2}$ | $80.1_{\pm 6.8}$ | $92.2_{\pm 1.5}$ | $70.7_{\pm 5.5}$ | $79.8_{\pm 3.8}$ |
| | S-CLIP | $81.4_{\pm 1.8}$ (+5.7) | $81.3_{\pm 3.4}$ (+1.2) | $95.9_{\pm 1.8}$ (+3.7) | $75.1_{\pm 2.0}$ (+4.4) | $86.4_{\pm 1.7}$ (+6.6) |
| | SEMICLIP | $\mathbf{83.8}_{\pm 1.0}$ (+8.1) | $\mathbf{85.6}_{\pm 3.4}$ (+5.5) | $\mathbf{96.6}_{\pm 1.2}$ (+4.4) | $\mathbf{75.3}_{\pm 1.9}$ (+4.6) | $\mathbf{87.4}_{\pm 0.5}$ (+7.6) |

Table 7: Zero-shot classification results on remote sensing datasets using different neural architectures. Parentheses indicate the performance gap from the supervised CLIP, and bolds denote the best results within each architecture.

## A.2    HYPERPARAMETER SENSITIVITY

Despite the impressive performance attained by our method, the inclusion of some parameters leads us to question whether these outstanding outcomes are a result of parameter tuning. Figure 3 presents the results of the parameter sensitivity experiments for $P$ (top-$P$% unlabeled images and surrogate captions are selected for training), $k$ (number of selected concepts for each unlabeled image), and $\mu$ (frequency threshold for concepts initialization). We observe that variations in $P$ and $k$ do not result in significant fluctuations in performance. However, when $\mu$ exceeds 5, the performance for zero-shot exhibits a noticeable decline. We believe this is due to a larger $\mu$ filtering out too many concepts, resulting in an insufficient number of concepts available for further consistency, which ultimately impacts model performance. Nevertheless, considering the strong performance of SEMICLIP across the majority of sensitivity experiments, it can be concluded that the performance advantage stems from its thoughtful design rather than an extensive reliance on parameter tuning.

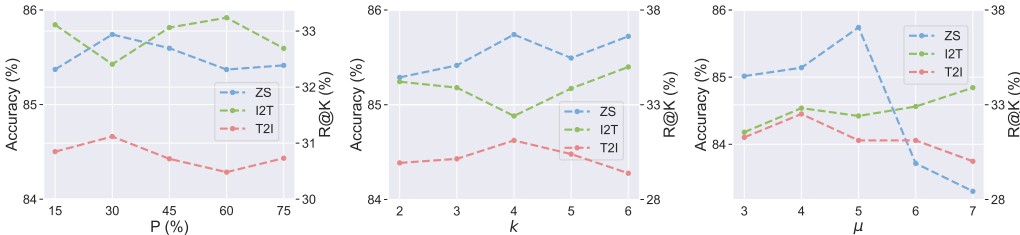

Figure 3: The sensitivity of P (%), $k$, and $\mu$ on remote sensing datasets. "ZS", "I2T", "T2I" denote the averaged zero-shot classification, image→text retrieval and text→image retrieval performance across various hyperparameter settings.

| Model | Method | Image→text R@5 | | | Text→image R@5 | | |
|---|---|---|---|---|---|---|---|
| | | RSICD | UCM | Sydney | RSICD | UCM | Sydney |
| ResNet-50 | CLIP (original) | 9.4 | 34.3 | 36.2 | 10.1 | 24.8 | 51.7 |
| | CLIP (fine-tuned) | $15.4_{\pm1.7}$ | $41.3_{\pm1.8}$ | $47.1_{\pm6.5}$ | $15.1_{\pm1.0}$ | $40.9_{\pm1.6}$ | $56.1_{\pm2.4}$ |
| | S-CLIP | $18.4_{\pm0.6}$ | $45.7_{\pm1.4}$ | $\mathbf{50.0}_{\pm3.0}$ | $16.8_{\pm1.2}$ | $43.5_{\pm1.5}$ | $55.1_{\pm2.0}$ |
| | SEMICLIP | $\mathbf{18.6}_{\pm1.9}$ | $\mathbf{46.2}_{\pm1.0}$ | $49.6_{\pm1.5}$ | $\mathbf{17.1}_{\pm0.7}$ | $\mathbf{44.3}_{\pm2.1}$ | $\mathbf{55.4}_{\pm1.8}$ |
| ViT-B-32 | CLIP (original) | 12.1 | 37.6 | 41.4 | 12.9 | 35.2 | 37.9 |
| | CLIP (fine-tuned) | $24.5_{\pm0.6}$ | $56.3_{\pm1.5}$ | $47.1_{\pm5.0}$ | $24.6_{\pm1.0}$ | $57.0_{\pm1.1}$ | $50.0_{\pm4.6}$ |
| | S-CLIP | $25.0_{\pm0.8}$ | $55.7_{\pm2.5}$ | $43.1_{\pm3.0}$ | $23.9_{\pm0.5}$ | $59.2_{\pm1.8}$ | $51.7_{\pm1.7}$ |
| | SEMICLIP | $\mathbf{26.7}_{\pm1.1}$ | $\mathbf{60.8}_{\pm1.8}$ | $\mathbf{48.9}_{\pm5.5}$ | $\mathbf{25.3}_{\pm1.0}$ | $\mathbf{60.0}_{\pm3.1}$ | $\mathbf{52.8}_{\pm2.6}$ |
| ViT-B-16 | CLIP (original) | 12.6 | 46.7 | 44.8 | 13.9 | 39.5 | 44.8 |
| | CLIP (fine-tuned) | $25.7_{\pm2.3}$ | $55.2_{\pm0.8}$ | $46.6_{\pm3.0}$ | $24.9_{\pm0.6}$ | $56.3_{\pm1.6}$ | $49.4_{\pm2.0}$ |
| | S-CLIP | $27.5_{\pm1.1}$ | $57.0_{\pm1.5}$ | $51.1_{\pm4.3}$ | $25.6_{\pm0.6}$ | $57.3_{\pm4.1}$ | $50.6_{\pm2.6}$ |
| | SEMICLIP | $\mathbf{27.6}_{\pm1.6}$ | $\mathbf{59.2}_{\pm2.2}$ | $\mathbf{53.4}_{\pm4.6}$ | $\mathbf{26.4}_{\pm0.4}$ | $\mathbf{58.4}_{\pm2.4}$ | $\mathbf{58.0}_{\pm2.0}$ |

Table 8: Image-text retrieval results on remote sensing datasets using different neural architecture. Bolds denote the best results within the same setups.

| Method | Data | Image→text R@1 | | | Text→image R@1 | | |
|---|---|---|---|---|---|---|---|
| | | RSICD | UCM | Sydney | RSICD | UCM | Sydney |
| CLIP (original) | - | 4.5 | 17.1 | 10.3 | 3.6 | 9.0 | 12.1 |
| CLIP (fine-tuned) | L | $10.6_{\pm1.2}$ | $18.9_{\pm2.7}$ | $20.7_{\pm0.0}$ | $7.3_{\pm0.7}$ | $14.9_{\pm0.5}$ | $12.6_{\pm1.0}$ |
| Hard-PL (Lee et al., 2013) | | $11.4_{\pm1.7}$ | $17.0_{\pm2.9}$ | $17.9_{\pm2.8}$ | $6.8_{\pm0.5}$ | $14.1_{\pm2.3}$ | $20.7_{\pm3.0}$ |
| Soft-PL (Assran et al., 2021) | L=U | $11.6_{\pm1.2}$ | $19.2_{\pm1.1}$ | $20.7_{\pm3.0}$ | $7.1_{\pm0.4}$ | $13.7_{\pm0.6}$ | $20.3_{\pm5.6}$ |
| S-CLIP (Mo et al., 2023) | | $11.4_{\pm0.3}$ | $19.5_{\pm1.3}$ | $17.8_{\pm2.6}$ | $7.1_{\pm0.8}$ | $14.0_{\pm1.4}$ | $16.7_{\pm1.0}$ |
| SEMICLIP (ours) | | $\mathbf{12.0}_{\pm0.6}$ | $\mathbf{20.3}_{\pm1.7}$ | $\mathbf{21.8}_{\pm4.0}$ | $\mathbf{7.6}_{\pm0.9}$ | $\mathbf{14.9}_{\pm2.8}$ | $\mathbf{21.3}_{\pm3.6}$ |
| Hard-PL (Lee et al., 2013) | | $11.1_{\pm1.5}$ | $17.1_{\pm2.4}$ | $19.0_{\pm3.4}$ | $6.1_{\pm0.2}$ | $16.0_{\pm1.5}$ | $16.7_{\pm5.3}$ |
| Soft-PL (Assran et al., 2021) | L≠U | $11.2_{\pm0.4}$ | $16.0_{\pm1.5}$ | $16.7_{\pm3.6}$ | $6.5_{\pm0.2}$ | $15.1_{\pm2.2}$ | $\mathbf{18.4}_{\pm1.0}$ |
| S-CLIP (Mo et al., 2023) | | $10.9_{\pm0.6}$ | $18.9_{\pm1.5}$ | $20.4_{\pm2.2}$ | $7.2_{\pm0.6}$ | $16.5_{\pm1.2}$ | $14.9_{\pm2.6}$ |
| SEMICLIP (ours) | | $\mathbf{11.5}_{\pm0.5}$ | $\mathbf{22.1}_{\pm0.5}$ | $20.7_{\pm3.4}$ | $\mathbf{7.3}_{\pm0.5}$ | $\mathbf{17.7}_{\pm1.7}$ | $17.8_{\pm4.3}$ |

Table 9: Image-text retrieval results of R@1 on remote sensing datasets, following the same setup of Table 1. Bolds denote the best results within the same setups.

## A.3 RESULTS FOR R@1

Tabs. 9 and 10 illustrate the R@1 performance of image-text retrieval for remote sensing and fashion datasets. For remote sensing datasets, SEMICLIP achieves performance advantages of 1.42% and 1.70% over S-CLIP in image→text and text→image retrieval, respectively. Similarly, on the fashion datasets, SEMICLIP achieves performance gains of 4.33% and 3.70% for image→text and text→image retrieval. It is evident that SEMICLIP achieves a more significant advantage on fashion datasets than remote sensing datasets, which we attribute to the larger sample size and greater diversity of the fashion datasets. This diversity makes it difficult for S-CLIP to identify suitable neighboring samples to generate effective pseudo-labels.

## A.4 RESULTS FOR SMALLER BATCH SIZE

All experiments in our main paper are conducted on four NVIDIA A6000 GPUs with a batch size of 64 per GPU. Given the potential limitations of computational resources, we conducted experiments to evaluate SEMICLIP on a single GPU, where the batch size is only one-quarter of that used in previous experiments. Tab. 11 reveals that SEMICLIP demonstrates a significant performance advantage over S-CLIP, achieving an average improvement of 7.54% in zero-shot classification. Notably, the performance of S-CLIP is even 1.72% lower than that of fine-tuning CLIP with only labeled image-text pairs, indicating its inability to achieve satisfactory performance with a smaller batch size due to the reliance on neighboring information within the batch and motivation under the paradigm of partial label learning (Xia et al., 2022; 2023). In contrast, SEMICLIP is less affected by batch size, consistently demonstrating robust improvements in the model's alignment capabilities.

| Method | Image→text R@1 | | | Text→image R@1 | | |
|---|---|---|---|---|---|---|
| | Fashion200k | FashionGen | Polyvore | Fashion200k | FashionGen | Polyvore |
| CLIP (original) | 4.7 | 10.9 | 11.4 | 4.5 | 12.5 | 13.0 |
| CLIP (fine-tuned) | $4.8_{\pm0.3}$ | $12.9_{\pm0.4}$ | $6.4_{\pm0.2}$ | $4.2_{\pm0.0}$ | $13.0_{\pm0.0}$ | $6.2_{\pm0.1}$ |
| Hard-PL (Lee et al., 2013) | $4.8_{\pm0.2}$ | $11.1_{\pm0.2}$ | $6.5_{\pm0.3}$ | $5.1_{\pm0.5}$ | $13.7_{\pm0.4}$ | $7.8_{\pm0.3}$ |
| Soft-PL (Assran et al., 2021) | $5.6_{\pm0.3}$ | $14.7_{\pm0.4}$ | $8.2_{\pm0.2}$ | $5.3_{\pm0.4}$ | $15.4_{\pm0.5}$ | $9.8_{\pm0.0}$ |
| S-CLIP (Mo et al., 2023) | $6.6_{\pm0.3}$ | $16.7_{\pm0.4}$ | $10.4_{\pm0.2}$ | $6.1_{\pm0.2}$ | $18.9_{\pm0.2}$ | $10.9_{\pm0.4}$ |
| SEMICLIP (ours) | $\mathbf{9.3}_{\pm0.3}$ | $\mathbf{24.0}_{\pm0.3}$ | $\mathbf{13.4}_{\pm0.5}$ | $\mathbf{8.7}_{\pm0.1}$ | $\mathbf{25.2}_{\pm0.5}$ | $\mathbf{13.1}_{\pm0.3}$ |

Table 10: Image-text retrieval results of R@1 on fashion datasets, following the same setup of Table 3. Bolds denote the best results within the same setups.

| Method | RSICD-CLS | UCM-CLS | WHU-RS19 | RSSCN7 | AID |
|---|---|---|---|---|---|
| CLIP (original) | 59.2 | 60.2 | 81.2 | 69.0 | 59.6 |
| CLIP (fine-tuned) | $75.3_{\pm2.7}$ | $77.2_{\pm7.6}$ | $92.3_{\pm2.9}$ | $72.4_{\pm3.2}$ | $79.7_{\pm3.5}$ |
| Hard-PL (Lee et al., 2013) | $78.5_{\pm3.3}$ (+3.2) | $70.1_{\pm4.7}$ ( -7.1) | $91.6_{\pm0.4}$ ( -0.7) | $62.4_{\pm4.8}$ ( -10.0) | $82.2_{\pm2.9}$ (+2.5) |
| Soft-PL (Assran et al., 2021) | $77.8_{\pm3.4}$ (+2.5) | $71.2_{\pm3.7}$ ( -6.0) | $94.3_{\pm1.0}$ (+2.0) | $64.9_{\pm5.5}$ ( -7.5) | $82.6_{\pm1.6}$ (+2.9) |
| S-CLIP (Mo et al., 2023) | $74.2_{\pm1.6}$ ( -1.1) | $69.7_{\pm3.0}$ ( -7.5) | $93.3_{\pm1.5}$ (+1.0) | $71.3_{\pm1.7}$ ( -1.1) | $79.8_{\pm1.6}$ (+0.1) |
| SEMICLIP (ours) | $\mathbf{83.5}_{\pm0.8}$ (+8.2) | $\mathbf{87.5}_{\pm3.8}$ (+10.3) | $\mathbf{95.1}_{\pm1.1}$ (+2.8) | $\mathbf{72.5}_{\pm3.1}$ (+0.1) | $\mathbf{87.4}_{\pm1.0}$ (+7.7) |

Table 11: Zero-shot classification results for batch size 64 of a single GPU on remote sensing datasets. We compare the original CLIP, supervised CLIP fine-tuned on labeled data, and semi-supervised methods that utilize unlabeled data sampled from the same distribution as the labeled data. Parentheses indicate the performance gap from the supervised CLIP, where values highlighted in green indicate gaps larger than one. Bolds denote the best results within the same setups.

## B COMPETING METHODS

We denote the pre-trained CLIP as CLIP (original) and the model fine-tuned only on image-text pairs as CLIP (fine-tuned). Following S-CLIP, we compare SEMICLIP with extensions of SSL methods, including Hard-PL and Soft-PL.

**Hard-PL.** For an unlabeled image $I_j$, Hard-PL searches for the labeled image within the same batch that is most similar to $I_j$ in terms of image embeddings and aligns its corresponding text with $I_j$ through Equation (1).

**Soft-PL.** Soft-PL employs soft nearest neighbor (Assran et al., 2021) to achieve alignment for unlabeled images. Unlike Hard-PL, Soft-PL takes into account the cosine similarities between unlabeled image $I_j$ and multiple nearest neighbors in labeled data, and applies the softmax function to obtain the soft pseudo-label which tends to have a value close to one for visually similar images. More details can be found in Assran et al. (2021); Mo et al. (2023).

**S-CLIP.** S-CLIP (Mo et al., 2023) features two innovative pseudo-labeling strategies: the caption-level pseudo-label combines captions from paired images by solving an optimal transport problem, while the keyword-level pseudo-label is derived from the nearest paired image's caption, employing partial label learning. It assumes a set of candidate labels for supervision instead of exact labels, enhancing contrastive learning and the integration of the language modality. S-CLIP and SEMICLIP are designed to improve performance by effectively utilizing additional unpaired data during training.

## C MORE EXPLANATION FOR CAPTION-LEVEL TRAPEZOIDAL CONSISTENCY

**Distinctions between CyCLIP and Caption-level Trapezoidal Consistency.** Considering that there exists some resemblance between CyCLIP and caption-level trapezoidal consistency (CTC), we delineate their differences and contributions of CTC below:

- CTC introduces CyCLIP into the semi-supervised learning scenario by utilizing unlabeled images and surrogate captions, thereby extending the applicability of CyCLIP beyond complete image-text pairs.
- We found that CTC can partially alleviate the impact of noise in the captions on training and provided a geometric explanation through trapezoidal geometric representation aspect.

- CTC offers new insights for future work addressing less-than-ideal alignment in image-text pairs, which is a widespread issue in practical settings.

**Why is CTC effective?** We explain the reasons why CTC is effective below:

- Traditional contrastive learning methods, such as CLIP, aim to reduce the distance between matching image-text pairs when learning image-text representations. However, they do not impose constraints on the overall geometric structure of the data, which leads to inconsistent predictions between the image and text spaces, especially in semi-supervised scenarios with only a small number of image-text pairs. Trapezoidal consistency addresses this issue by introducing equal-length legs and diagonals consistency regularization terms. These regularizers constrain the similarity gaps between mismatched image-text pairs as well as image-image and text-text pairs, resulting in a more consistent and structured representation space, thereby improving prediction consistency.
- Trapezoidal consistency can ensure geometric alignment between image and text representations, allowing the model to make more consistent predictions when reasoning in both the image and text spaces. This means that the image and text representations learned by trapezoidal consistency can be more easily interchanged, leading to improved performance on downstream tasks.
- The rigid separation between the positive pairs and negative pairs enforced by the contrastive loss in CLIP may degrade performance when some pairs in the negative batch belong to a similar entity (Mu et al., 2022). Trapezoidal consistency poses constraints on the overall geometry of all the data pairs rather than forcing a rigid separation, which enables the interaction of information between intra-modal and cross-modal samples, even in data-scarce scenarios.

## D  MORE EXAMPLES FOR PREDICTED CONCEPTS

Figure 4 presents additional examples of predicted concepts for unlabeled images on remote sensing and fashion datasets. We observe that although some concepts do not explicitly appear in the ground-truth captions, they are still highly semantically relevant to the unlabeled images, as illustrated by "stadium" in Figure 4a and "midi" in Figure 4d. The strong semantic alignment of the extracted concepts with the images provides a solid foundation for the subsequent concept-level semantic consistency and caption-level trapezoidal consistency in proposed SEMICLIP.

## E  WHY SEMICLIP CAN OUTPERFORM ORACLE IN SOME SETTINGS?

Interestingly, we find that our method can outperforms the oracle in some zero-shot settings. We believe this is due to the severe imbalance in the proportions of the sub-datasets within the training sets, RS-ALL and Fashion. When training on the full dataset, sub-datasets with a smaller proportion, such as UCM in Tab. 1, perform poorly on their corresponding zero-shot test set(UCM-CLS), which resulted in a slightly lower overall performance. We find this to be an interesting phenomenon, and it warrants further research into how to mitigate the issue of imbalance in the proportions of the sub-datasets.

The oracle outperforms SEMICLIP in retrieval performance, indicating that increasing the amount of labeled data will significantly improve retrieval performance. This also points to a potential direction for further improvement of SEMICLIP in the future.

## F  THE SIZE OF DATASETS

For remote sensing datasets, the RSICD, UCM, and Sydney datasets contain 8734, 1680, and 497 image-text pairs, respectively. The three datasets constitute RS-ALL, and we randomly select 10% of the image-text pairs from the training set as labeled data, with the remaining pairs treated as unlabeled. During the inference, the sizes for classification datasets are shown in Tab. 12.

For fashion datasets, the Fashion200k, FashionGen, and Polyvore datasets contain 61753, 60147, and 71967 image-text pairs, respectively. The sizes for classification datasets are shown in Tab. 13.

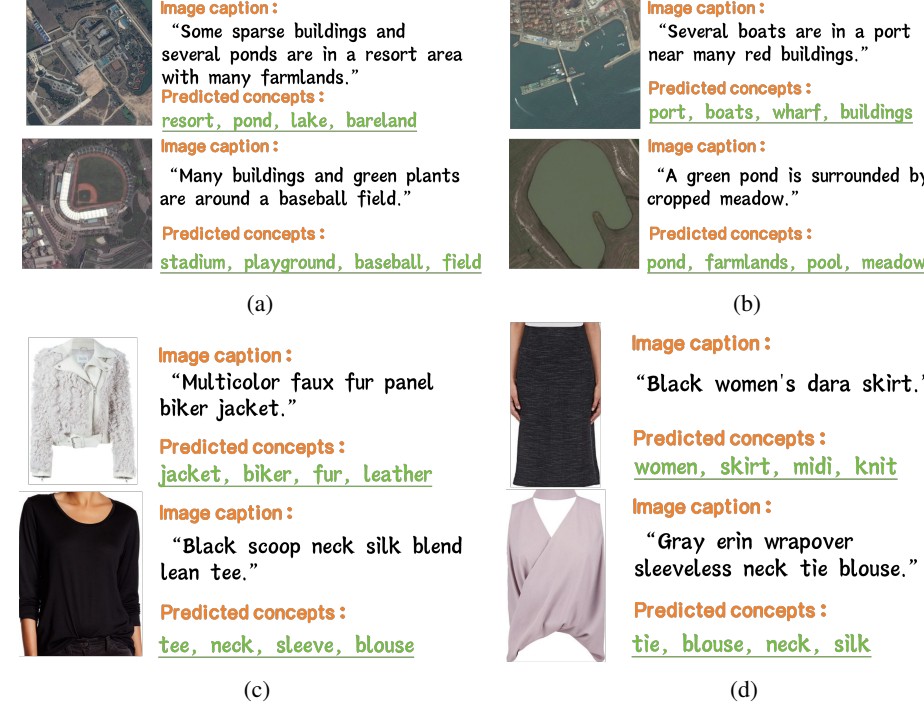

Figure 4: (4a and 4b) More examples for predicted concepts for unlabeled images in remote sensing datasets; (4c and 4d) More examples for predicted concepts for unlabeled images in fashion datasets.

Table 12: The size of zero-shot classification datasets in the remote sensing domain.

|  | RSICD-CLS | UCM-CLS | WHU-RS19 | RSSCN7 | AID |
|---|---|---|---|---|---|
| Number of images | 1094 | 2100 | 1005 | 2800 | 10000 |
| Number of classes | 31 | 21 | 19 | 7 | 30 |

Table 13: The size of zero-shot classification datasets in the fashion domain.

|  | Fashion200k | | FashionGen | | Polyvore |
|---|---|---|---|---|---|
|  | Super-class | Sub-class | Super-class | Sub-class | Class |
| Number of images | 29,785 | | 32,528 | | 14,657 |
| Number of classes | 5 | 31 | 48 | 121 | 11 |

For other datasets, SciCap contains 106934 image-text pairs and Simpsons only contains 720 pairs.

## G  THE COMPARISON WITH NLIP

Due to concept labeler in Nlip (Huang et al., 2023) looks similar to semantic concepts mining, the differences are as follows:

- We train a linear classifier to achieve better image-concept alignment in task-specific domain, while concept labeler in Huang et al. (2023) obtains concepts through image-concept retrieval, which may not enable effective retrieval in certain specialized domains due to the niche nature of some concepts.
- Concepts generated in Nlip are considered as conditional input to the cross-modal decoder, which requires a large amount of data for training, significantly increasing the overhead. However, SEMICLIP can be trained with only a small amount of labeled data, achieving significant improvements in model performance in semi-supervised settings.

## H    THE ANALYSIS OF PROMPTING STRATEGY

We analyze the prompting strategy experimentally for a more in-depth investigation.

**Comparing performance with prompts at different positions.** We conducted experiments on the effect of prompts at different positions, with all prompts initialized using a normal distribution for fair comparison. Overall, as shown in Tab. 14, the performance is slightly better when the prompts are positioned in the middle, which is why we use this approach in our experiments.

Table 14: Effects of incorporating prompts at various positions.

| Position | Beginning | | | Middle | | | End | | |
|---|---|---|---|---|---|---|---|---|---|
| | ZS | I2T | T2I | ZS | I2T | T2I | ZS | I2T | T2I |
| Accuracy or Recall | 84.2 | 32.2 | 30.9 | 84.9 | 32.3 | 30.6 | 84.4 | 31.7 | 30.3 |

**Different Prompts initialization.** From the results in Tab. 15, we found that zero initialization performed poorly in zero-shot tasks, with 1.1% lower performance compared to SEMICLIP, but showed no significant decline in retrieval performance. Normal initialization performs slightly worse than our proposed method in both zero-shot and retrieval tasks. Overall, SEMICLIP's initialization method achieves the most robust and stable performance.

Table 15: Effects of incorporating prompts of different initialization.

| Initialization | Zero | | | Normal | | | SemiCLIP | | |
|---|---|---|---|---|---|---|---|---|---|
| | ZS | I2T | T2I | ZS | I2T | T2I | ZS | I2T | T2I |
| Accuracy or Recall | 84.6 | 32.5 | 31.0 | 84.9 | 32.3 | 30.6 | 85.7 | 32.4 | 31.1 |

**Fixed vs learnable prompts.** The results in Tab. 6 show that learnable prompts achieve an average performance advantage of 0.9% over fixed prompts, which indicates that learnable prompts facilitate improved model adaptation to task-specific domains.

## I    EFFECTS OF DIFFERENT LABELED DATA PERCENTAGES

The experimental results in Figure 5 show that SEMICLIP consistently outperforms S-CLIP across different label proportion settings. In settings with a low proportion of labeled data, the quality of extracted concepts is impacted. However, methods like S-CLIP, which rely on labeled data to construct neighbor labels, will experience a more significant performance drop in such scenarios. The above experiments demonstrate the robustness of the proposed training framework under different proportions of labeled data.

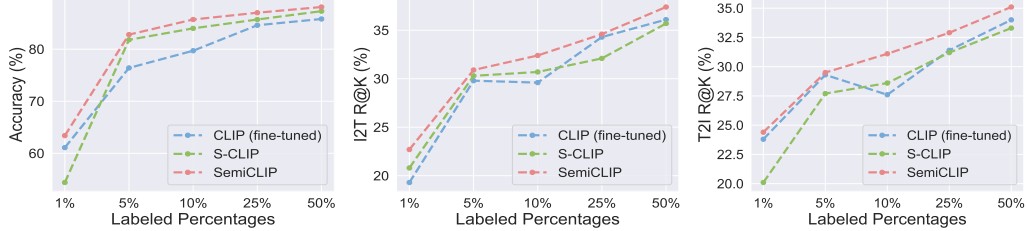

Figure 5: Ablation study regarding different percentages of labeled data.

## J    EXPERIMENTS ON GENERAL BENCHMARKS

To evaluate the performance of SEMICLIP on general benchmarks, we conducted comparative experiments on the COCO (Lin et al., 2014) dataset. The results in Tab. 16 indicate that SEMICLIP

| Method | 1% labeled | | | | 10% labeled | | | |
| | Image→text | | Text→image | | Image→text | | Text→image | |
| | R@1 | R@5 | R@1 | R@5 | R@1 | R@5 | R@1 | R@5 |
|---|---|---|---|---|---|---|---|---|
| CLIP (fine-tuned) | $33.7_{\pm1.3}$ | $60.2_{\pm0.9}$ | $33.6_{\pm1.1}$ | $60.5_{\pm2.0}$ | $35.6_{\pm1.2}$ | $65.0_{\pm2.8}$ | $36.9_{\pm1.5}$ | $65.0_{\pm1.3}$ |
| S-CLIP (Mo et al., 2023) | $29.5_{\pm1.8}$ | $54.8_{\pm0.9}$ | $27.8_{\pm1.4}$ | $53.8_{\pm1.1}$ | $33.2_{\pm1.2}$ | $60.6_{\pm3.1}$ | $31.7_{\pm1.6}$ | $59.1_{\pm1.0}$ |
| SEMICLIP (ours) | $\mathbf{37.8}_{\pm1.2}$ | $\mathbf{64.4}_{\pm3.1}$ | $\mathbf{37.6}_{\pm1.2}$ | $\mathbf{65.7}_{\pm3.8}$ | $\mathbf{41.4}_{\pm1.3}$ | $\mathbf{70.3}_{\pm1.2}$ | $\mathbf{42.6}_{\pm2.9}$ | $\mathbf{69.8}_{\pm2.4}$ |

Table 16: The retrieval performance on COCO.

can achieve significant performance improvements on general benchmark over CLIP (fine-tuned) and S-CLIP. It is worth noting that S-CLIP's performance shows an average decrease of 4.5% compared to CLIP (fine-tuned), aligning with the paper's claim (Mo et al., 2023) that S-CLIP experiences performance drops when trained on a small number of image-text pairs in common datasets like COCO. However, the superior performance of our proposed SEMICLIP is unaffected by the different types of datasets, achieving significant improvements on both commonly used datasets and task-specific datasets.

