# OpenReview forum: "Semi-Supervised CLIP Adaptation by Enforcing Semantic and Trapezoidal Consistency"
_ICLR.cc/2025/Conference — ICLR 2025 Poster_

### Official Review · Reviewer_JQns · 2024-10-16

**Soundness:** 3
**Presentation:** 3
**Contribution:** 3
**Rating:** 6
**Confidence:** 4

**Summary:**

The paper proposes a semi-supervised CLIP training method, termed SemiCLIP, which leverages a large amount of unlabeled images when only limited image-text pairs are available. SemiCLIP consists of two training stages: supervised pre-training and semi-supervised fine-tuning. In the supervised pre-training stage, the paper introduces a concept classifier along with the standard contrastive loss to enhance image-text alignment using labeled data. The semi-supervised fine-tuning stage includes two key components: concept-level semantic consistency, which ensures that the model maintains consistency in understanding visual concepts, and caption-level trapezoidal consistency, designed to improve cross-modal alignment by refining the geometric structure between image-text pairs. Experimental results show promising improvements over existing baselines across diverse domains.

**Strengths:**

- The paper is well written and easy to follow.
- Experiments are comprehensive.
- Results show consistent improvements over baselines across diverse domains.

**Weaknesses:**

The motivation behind caption-level trapezoidal consistency (Section 3.3.2) is not sufficiently explained.

- The surrogate caption is generated by concatenation of templated keywords. Even though the templates are learnable, the distribution of generated captions is likely to differ significantly from the labeled captions. It would be helpful if the underlying motivation for this choice of surrogate caption generation process is explained in detail. (One simple alternative might be generating surrogate captions based on keywords using a large language model.)
- The main objective of trapezoidal consistency loss is to effectively utilize less-than-ideal surrogate captions due to their inexact nature. However, it is unclear why diagonal and legs regularization is beneficial, whereas the direct usage of contrastive loss is not, given the same inexact caption. It would be useful to explain why trapezoidal consistency is better suited to handle noisy captions. (A simpler alternative could involve using a soft-label in the contrastive loss, specifically for surrogate captions.)

I look forward to authors’ clarification and am willing to increase the score based on their clarification.

**Questions:**

- Effect of linear classifier: Semantic concept classifier could be performed by CLIP text encoder, akin to zero-shot classification. Is there a specific reason for using linear classifier head instead?

---

> ### Author Response · Authors · 2024-11-16
>
> Dear Reviewer JQns,
>
> We deeply appreciate the reviewer's thoughtful comments. We are encouraged by comments such as *well written* and *experiments are comprehensive*. We will respond to each of your concerns below.
>
> **Weakness #1: The underlying motivation for surrogate caption generation process.**
>
> **Response:** Following your suggestion, we generated captions based on keywords using a large language model (GPT-4o). The comparison results are as follows:
>
> |  Remote sensing   |  ZS  | I2T  | T2I  |
> | :---------------: | :--: | :--: | :--: |
> |     SemiCLIP      | 85.7 | 32.4 | 31.1 |
> | SemiCLIP (GPT-4o) | 84.9 | 33.3 | 30.0 |
>
> In the table, we report the average performance of ZS (zero-shot), I2T (image→text retrieval), and T2I (text→image retrieval) on the remote sensing datasets. From the results, we found that the surrogate captions generated by SemiCLIP generally perform better than the captions generated by GPT-4o based on keywords. We believe this is due to the lack of relationships between concepts, which makes it difficult to reconstruct a complete caption based solely on keywords. As a result, the captions generated by GPT-4o still contain a significant amount of noise.
>
> The surrogate captions in SemiCLIP are generated by the concatenation of keywords and learnable prompts, where learnable prompts can help surrogate captions better adapt to the task-specific domains, leading to improved performance. In fact, generating captions for images is a challenging task in scenarios with limited data and models. The surrogate captions we proposed offer a simple solution to this task, with hopes for better methods in the future.
>
> **Weakness #2: Why diagonal and legs regularization are beneficial?**
>
> **Response:** We first explain the reasons why diagonal and legs regularization are beneficial below:
>
> (1) Traditional contrastive learning methods, such as CLIP, aim to reduce the distance between matching image-text pairs when learning image-text representations. However, they do not impose constraints on the overall geometric structure of the data, which leads to **inconsistent predictions between the image and text spaces**, especially in semi-supervised scenarios with only a small number of image-text pairs. Trapezoidal consistency addresses this issue by introducing equal-length legs and diagonals consistency regularization terms. These regularizers constrain the similarity gaps between mismatched image-text pairs as well as image-image and text-text pairs, resulting in a more consistent and structured representation space, thereby improving prediction consistency.
>
> (2) Trapezoidal consistency can ensure geometric alignment between image and text representations, allowing the model to make more consistent predictions when reasoning in both the image and text spaces. This means that the image and text representations learned by trapezoidal consistency can be **more easily interchanged**, leading to improved performance on downstream tasks.
>
> (3) The **rigid separation** between the positive pairs and negative pairs enforced by the contrastive loss in CLIP may degrade performance when some pairs in the negative batch belong to a similar entity. Trapezoidal consistency poses constraints on the overall geometry of all the data pairs rather than forcing a rigid separation, which enables the **interaction** of information between intra-modal and cross-modal samples, even in data-scarce scenarios.
>
> Based on your suggestion, we also conducted relevant experiments using a soft-label approach, and the results are as follows:
>
> |    Remote sensing    |  ZS  | I2T  | T2I  |
> | :------------------: | :--: | :--: | :--: |
> |       SemiCLIP       | 85.7 | 32.4 | 31.1 |
> | Soft-label version 1 | 82.1 | 31.7 | 30.8 |
> | Soft-label version 2 | 82.7 | 31.0 | 29.9 |
>
> The soft-label version 1 refers to the use of similarity between unlabeled images and surrogate captions to directly weight the contrastive loss applied to them, with the aim of mitigating the impact of noise in the surrogate captions. The soft-label version 2 is the Soft-PL, which employs soft nearest neighbor to achieve alignment for unlabeled images and has been compared in most tables in our paper.
>
> The results indicate that the soft-label methods fail to effectively mitigate the noise in the captions under this scenario. However, our proposed diagonal and legs regularization, through indirect constraints on the unlabeled images and surrogate captions, effectively enhances the model's alignment performance.

---

> > ### Author Response · Authors · 2024-11-16
> >
> > **Question #1: Effect of linear classifier.**
> >
> > **Response:** We answer this issue by developing an open-set classifier. Specifically, through a stage of CLIP loss training with labeled data, we leverage the model's **zero-shot capabilities** to assign corresponding concepts to each image, which performs the role of an open-set classifier. The concepts list is still extracted from the captions of the labeled data, and for each image, we select the concepts with top-k cosine similarity. The second stage is consistent with SemiCLIP, except that the concepts are generated by open-set classifier. The experimental results of averaged performance on remote sensing and fashion datatsets are as follows:
> >
> > |         Remote sensing          |  ZS  | I2T  | T2I  |
> > | :-----------------------------: | :--: | :--: | :--: |
> > | SemiCLIP (Close-set classifier) | 85.7 | 32.4 | 31.1 |
> > | SemiCLIP (Open-set classifier)  | 82.6 | 32.0 | 30.8 |
> >
> > |             Fashion             |  ZS  | I2T  | T2I  |
> > | :-----------------------------: | :--: | :--: | :--: |
> > | SemiCLIP (Close-set classifier) | 60.8 | 24.2 | 24.5 |
> > | SemiCLIP (Open-set classifier)  | 56.9 | 23.5 | 23.6 |
> >
> > The results reveal a performance decline when using the open-set classifier. We attribute this to the fact that some concepts in the task-specific domain are not well captured by the image-text alignment model, making it difficult to effectively generate the corresponding concepts. We noticed a significant decline in performance of zero-shot, with a **drop of 3.5%**. This suggests that close-set classifier is more robust to these task-specific concepts due to its image-concept alignment, resulting in better performance.

---

> ### Comment · Reviewer_JQns · 2024-11-20
>
> I appreciate authors’ additional experiment and clarification, which resolve my concerns. I update the rating accordingly.

---

> > ### Author Response · Authors · 2024-11-20
> >
> > Thank you for your valuable feedback and for recognizing our additional efforts. We greatly appreciate your time and support in improving our work.

---

### Official Review · Reviewer_Psom · 2024-10-18

**Soundness:** 3
**Presentation:** 3
**Contribution:** 2
**Rating:** 6
**Confidence:** 3

**Summary:**

In this manuscript, the authors focus on efficiently mitigating the domain gap in semi-supervised learning of CLIP on downstream tasks. Specifically, the authors first leverage labeled images to extract the concept list (used as pseudo labels) of the target dataset, and employ CLIP loss and classification loss on pseudo labels for supervised learning. After initialization on labeled set, the authors propose concept-level semantic consistency and caption-level trapezoidal consistency to optimize CLIP on the whole dataset. Experimental results show that the proposed method can effectively

**Strengths:**

1. The correlation between images and concepts is intuitive, which can be effectively used to optimize CLIP in semi-supervised learning scenarios.

2. The experimental results are promising.

3. The writing is polished and easy to understand.

**Weaknesses:**

1. My main concern is the analysis of prompting strategy of \hat{T}_{j}. The position of prompts, the initialization of prompts, or even using fixed prompts rather than learnable prompts can be analyzed. The authors can conduct empirical analysis to investigate which prompting strategy is better. For example, comparing performance with prompts at different positions (e.g. beginning, middle (similar to current configuration), and end of all concepts), different initialization (zero init or normal init for prompt), and fixed vs learnable prompts. All these factors may affect the quality of pseudo captions for unlabeled images.

2. The quality of pseudo labels (i.e., mined semantic concepts) may be critical for semi-supervised learning. The authors could conduct some analysis (e.g., mining concepts with different granularities) to investigate the effect from quality of pseudo labels. For example, "your current configuration" vs "mined concepts but unified into some common objects or attributes" vs "oracle fine-grained concepts (e.g., mining concepts from the whole original dataset)".

3. The authors can also conduct ablation study regarding different percentages of labeled data (e.g. 1%, 5%, 10%, 25%, 50% labeled data) to evaluate the robustness of proposed training framework. Intuitively, a good semi-supervised learning framework may still robust against low percentage labeled data. Nevertheless, in low percentage labeled data settings, the lack of ground-truth caption may restrict the quality of mined concepts.

**Questions:**

For Table 1~4, we recommend  the authors to show the upper bound (supervised learning on all data) to reveal the gap between semi-supervised learning and fully-supervised learning.

---

> ### Author Response · Authors · 2024-11-16
>
> Dear Reviewer Psom,
>
> We are grateful for the thoughtful reviews, and for the comments like *effectively*, *results are promising*, and *easy to understand*. We will address the concerns below.
>
> **Weakness #1: The analysis of prompting strategy.**
>
> **Response:** Thank you for your valuable suggestions to analyze the prompting strategy experimentally. We believe that a more in-depth investigation is indeed necessary. Based on your advice, we carried out the corresponding experiments, and report the average performance of ZS (zero-shot), I2T (image→text retrieval), and T2I (text→image retrieval) on the remote sensing datasets below:
>
> (1) **Comparing performance with prompts at different positions.**
>
> |   Remote sensing   |  ZS  | I2T  | T2I  |
> | :----------------: | :--: | :--: | :--: |
> | Position beginning | 84.2 | 32.2 | 30.9 |
> |  Position middle   | 84.9 | 32.3 | 30.6 |
> |    Position end    | 84.4 | 31.7 | 30.3 |
>
> We conducted experiments on the effect of prompts at different positions, with all prompts initialized using a normal distribution. Overall, the performance is slightly better when the prompts are positioned in the middle, which is why we use this approach in our experiments.
>
> (2) **Different initialization.**
>
> |    Remote sensing     |  ZS  | I2T  | T2I  |
> | :-------------------: | :--: | :--: | :--: |
> |  Zero initialization  | 84.6 | 32.5 | 31.0 |
> | Normal initialization | 84.9 | 32.3 | 30.6 |
> |       SemiCLIP        | 85.7 | 32.4 | 31.1 |
>
> From the results above, we found that zero initialization performed poorly in zero-shot tasks, with 1.1% lower performance compared to SemiCLIP, but showed no significant decline in retrieval performance. Normal initialization performs slightly worse than our proposed method in both zero-shot and retrieval tasks. Overall, SemiCLIP's initialization method achieves the most robust and stable performance.
>
> (3) **Fixed vs learnable prompts.**
>
> In fact, we have presented the relevant results in the ablation study, specifically in Table 6, and we have reorganized the results as follows:
>
> |  Remote sensing   |  ZS  | I2T  | T2I  |
> | :---------------: | :--: | :--: | :--: |
> |   Fixed prompts   | 84.6 | 31.5 | 30.5 |
> | Learnable prompts | 85.7 | 32.4 | 31.1 |
>
> The results show that learnable prompts achieve an average performance advantage of 0.9% over fixed prompts.
>
> The experiments above demonstrate the effectiveness of the prompt strategy in SemiCLIP, and we will include the relevant results in the revised version of the paper.
>
> **Weakness #2: The quality of pseudo labels.**
>
> **Response:** We appreciate your insightful suggestions regarding the quality of pseudo labels, and the experimental results based on your advice are as follows:
>
> |        Remote sensing        |  ZS  | I2T  | T2I  |
> | :--------------------------: | :--: | :--: | :--: |
> |           SemiCLIP           | 85.7 | 32.4 | 31.1 |
> |       Common concepts        | 83.9 | 32.2 | 30.3 |
> | Oracle fine-grained concepts | 86.2 | 33.6 | 31.8 |
>
> For Common concepts, we use the class names that truly exist in the zero-shot test set of remote sensing as the concepts. For oracle fine-grained concepts, we extract the concepts from the real captions corresponding to the unlabeled images.
>
> From the results, we can see that SemiCLIP achieves an average performance improvement of 0.9% compared to using common concepts, while using oracle fine-grained concepts provides an additional 0.8% improvement over SemiCLIP. This suggests that finer-grained concepts may have a more positive impact on performance. However, in practice, oracle fine-grained concepts are not accessible. SemiCLIP effectively leverages predicted concepts to achieve performance close to that of oracle fine-grained concepts.

---

> > ### Author Response · Authors · 2024-11-16
> >
> > **Weakness #3: Ablation study regarding different percentages of labeled data.**
> >
> > **Response:** We conduct ablation study regarding different percentages of labeled data, and the results are below:
> >
> > |    1% labeled     |  ZS  | I2T  | T2I  |
> > | :---------------: | :--: | :--: | :--: |
> > | CLIP (fine-tuned) | 61.1 | 19.3 | 23.8 |
> > |      S-CLIP       | 54.4 | 20.8 | 20.1 |
> > |     SemiCLIP      | 63.4 | 22.7 | 24.4 |
> >
> > |    5% labeled     |  ZS  | I2T  | T2I  |
> > | :---------------: | :--: | :--: | :--: |
> > | CLIP (fine-tuned) | 76.4 | 29.8 | 29.3 |
> > |      S-CLIP       | 81.8 | 30.3 | 27.7 |
> > |     SemiCLIP      | 82.8 | 30.9 | 29.5 |
> >
> > |    10% labeled    |  ZS  | I2T  | T2I  |
> > | :---------------: | :--: | :--: | :--: |
> > | CLIP (fine-tuned) | 79.7 | 29.6 | 27.6 |
> > |      S-CLIP       | 84.0 | 30.7 | 28.6 |
> > |     SemiCLIP      | 85.7 | 32.4 | 31.1 |
> >
> > |    25% labeled    |  ZS  | I2T  | T2I  |
> > | :---------------: | :--: | :--: | :--: |
> > | CLIP (fine-tuned) | 84.6 | 34.3 | 31.4 |
> > |      S-CLIP       | 85.7 | 32.1 | 31.2 |
> > |     SemiCLIP      | 87.0 | 34.6 | 32.9 |
> >
> > |    50% labeled    |  ZS  | I2T  | T2I  |
> > | :---------------: | :--: | :--: | :--: |
> > | CLIP (fine-tuned) | 85.8 | 36.1 | 34.0 |
> > |      S-CLIP       | 87.3 | 35.7 | 33.3 |
> > |     SemiCLIP      | 88.1 | 37.4 | 35.1 |
> >
> > CLIP (fine-tuned) refers to CLIP fine-tuned using only labeled data. In the table, we report the average performance of ZS (zero-shot), I2T (image→text retrieval), and T2I (text→image retrieval) on the remote sensing datasets. The experimental results show that SemiCLIP consistently outperforms S-CLIP across different label proportion settings. In settings with a low proportion of labeled data, the quality of extracted concepts is indeed impacted. However, methods like S-CLIP, which rely on labeled data to construct neighbor labels, will experience a more significant performance drop in such scenarios. The above experiments demonstrate the robustness of the proposed training framework under different proportions of labeled data.
> >
> > **Question #1: The upper bound regarding supervised learning on all data.**
> >
> > **Response:** We show the averaged upper bound which we refer it as Oracle (fully supervised fine-tuned) below:
> >
> > |            Remote sensing            |  ZS  | I2T  | T2I  |
> > | :----------------------------------: | :--: | :--: | :--: |
> > |               SemiCLIP               | 85.7 | 32.4 | 31.1 |
> > | Oracle (fully supervised fine-tuned) | 82.0 | 39.9 | 36.6 |
> >
> > |               Fashion                |  ZS  | I2T  | T2I  |
> > | :----------------------------------: | :--: | :--: | :--: |
> > |               SemiCLIP               | 60.8 | 24.2 | 24.5 |
> > | Oracle (fully supervised fine-tuned) | 58.5 | 37.4 | 37.0 |
> >
> > Interestingly, our method even outperforms the oracle calculated here in zero-shot performance. We believe this is due to the severe **imbalance in the proportions of the sub-datasets** within the training sets, RS-ALL and Fashion. When training on the full dataset, sub-datasets with a smaller proportion, such as UCM, perform poorly on their corresponding zero-shot test set, UCM-CLS, which resulted in a slightly lower overall performance. We find this to be an interesting phenomenon, and it warrants further research into how to mitigate the issue of imbalance in the proportions of the sub-datasets.
> >
> > The oracle outperforms SemiCLIP in retrieval performance, indicating that increasing the amount of labeled data will significantly improve retrieval performance. This also points to a potential direction for further improvement of SemiCLIP in the future. We will include the oracle results in the experimental tables of the paper, helping readers gain a clearer understanding of the task.

---

> ### Author Response · Authors · 2024-11-25
>
> Dear Reviewer Psom,
>
> We sincerely thank the reviewer for valuable comments. We have addressed them in our responses and updated the manuscript accordingly. If the reviewer has any further questions, we are always ready to provide additional clarifications.
>
> Best regards,
>
> The Authors

---

> > ### Comment · Reviewer_Psom · 2024-12-01
> >
> > This response has addressed my concerns. I tend to accept this paper.

---

> > > ### Author Response · Authors · 2024-12-01
> > >
> > > Thank you for your feedback, and we appreciate your recommendation to accept the paper.

---

### Official Review · Reviewer_52fH · 2024-11-03

**Soundness:** 2
**Presentation:** 3
**Contribution:** 2
**Rating:** 6
**Confidence:** 4

**Summary:**

This paper presents a new semi-supervised training method for vision-language pre-training models like CLIP, called SemiCLIP. The method is designed to improve CLIP's adaptability to downstream tasks when there's limited image-text paired data. SemiCLIP uses a small amount of image-text pairs and a large volume of images without text descriptions to enhance cross-modal alignment. It introduces semantic concept mining to improve visual representations by matching images with relevant concepts from labeled data. The method also creates learnable surrogate captions for unlabeled images and optimizes a trapezoidal consistency to regulate the geometric structure of image-text pairs. The experiments show that SemiCLIP significantly improves CLIP's adaptability, increasing zero-shot classification accuracy by 1.72% - 6.58% and image-text retrieval performance by 2.32% - 3.23% on standard benchmarks.

**Strengths:**

+ The paper proposes a new method to leverage unlabeled images alongside limited labeled data, enhancing CLIP's cross-modal alignment.

+ Authors design a caption-level trapezoidal consistency loss to appropriately aggregate mined concepts, which is new to me.

**Weaknesses:**

- The method used to mine semantice concepts had been widely used in semi-supervised works thus is not quite novel to me.
- Why does the distance between I_i and Tˆ_j should be consistent with the distance between I_j and T_i? The reason for the consistency for diagonals constrains is not clear to me.

**Questions:**

1. Since the paper mainly focus on the CLIP downstream adaptation, I suggest authors change the title from "SEMI-SUPERVISED CLIP TRAINING" to "SEMI-SUPERVISED CLIP ADAPTING".

2. More explanation towards trapezoidal distance hypothesis. Why it is necessary to restrict the trapezoid to have equal-length legs and diagonals?

---

> ### Author Response · Authors · 2024-11-16
>
> Dear Reviewer 52fH,
>
> We are grateful for the valuable reviews, and for the positive comments such as *new method*. We will address the concerns below.
>
> **Weakness #1: The method used to mine semantic concepts is not novel.**
>
> **Response:** During the semantic concepts mining (SCM), the model is **adapted to a task-specific domain** and trained with a classifier that **aligns with the concepts in the images**, which lays an important foundation for generating surrogate captions in the second stage. We believe SCM is an **innovative approach to utilizing unlabeled data**, and significant improvement in model performance is achieved through image-concept alignment in Figure (2a) and Figure (2b).
>
> In fact, mining semantic concepts is not the primary innovation of this paper, and it mainly serves to subsequent generation of surrogate captions and further performance improvement via consistency loss. The improvement in mining semantic concepts does contribute to the performance enhancement for the problem addressed in this paper, but it is not the primary factor. Therefore, we chose to keep its design simple, allowing for further improvements in future research.
>
> **Weakness #2 and Question #2: Why it is necessary to restrict the trapezoid to have equal-length legs and diagonals?**
>
> **Response:** We will explain the reasons why it is necessary to restrict the trapezoid to have equal-length legs and diagonals below:
>
> (1) Traditional contrastive learning methods, such as CLIP, aim to reduce the distance between matching image-text pairs when learning image-text representations. However, they do not impose constraints on the overall geometric structure of the data, which leads to **inconsistent predictions between the image and text spaces**, especially in semi-supervised scenarios with only a small number of image-text pairs. Trapezoidal consistency addresses this issue by introducing equal-length legs and diagonals consistency regularization terms. These regularizers constrain the similarity gaps between mismatched image-text pairs as well as image-image and text-text pairs, resulting in a more consistent and structured representation space, thereby improving prediction consistency.
>
> (2) Trapezoidal consistency can ensure geometric alignment between image and text representations, allowing the model to make more consistent predictions when reasoning in both the image and text spaces. This means that the image and text representations learned by trapezoidal consistency can be **more easily interchanged**, leading to improved performance on downstream tasks.
>
> (3) The **rigid separation** between the positive pairs and negative pairs enforced by the contrastive loss in CLIP may degrade performance when some pairs in the negative batch belong to a similar entity. Trapezoidal consistency poses constraints on the overall geometry of all the data pairs rather than forcing a rigid separation, which enables the **interaction** of information between intra-modal and cross-modal samples, even in data-scarce scenarios.
>
> **Question #1: Suggestions for improving the paper title.**
>
> **Response:** Thank you for your suggestion! We will carefully consider your suggestion and make some revisions to the title.

---

> > ### Comment · Reviewer_52fH · 2024-11-27
> >
> > Thanks for your feedback. The explanation is technically sound to me thus I will raise my rating.

---

> > > ### Author Response · Authors · 2024-11-27
> > >
> > > We're glad to hear that the explanation is technically sound, and we appreciate your decision to raise your rating.

---

> ### Author Response · Authors · 2024-11-25
>
> Dear Reviewer 52fH,
>
> We sincerely thank the reviewer for valuable comments. We have addressed them in our responses and updated the manuscript accordingly. If the reviewer has any further questions, we are always ready to provide additional clarifications.
>
> Best regards,
>
> The Authors

---

### Official Review · Reviewer_WcrK · 2024-11-04

**Soundness:** 2
**Presentation:** 3
**Contribution:** 2
**Rating:** 5
**Confidence:** 4

**Summary:**

The paper presents SEMICLIP, a semi-supervised training method that enhances CLIP’s performance with limited number of image-text paired data. It utilizes a small number of labeled pairs along with a large set of unlabeled images by employing semantic concept mining to create pseudo-labels for the unlabeled data. The method introduces trapezoidal consistency regularization to maintain geometric relationships between image-text pairs, optimizing the model's alignment. Experimental results show that SEMICLIP bring improvements on zero-shot classification and image-text retrieval performance on various domains datasets.

**Strengths:**

- The paper is written in a clear manner, making the concepts and methodologies easy to understand. This clarity enhances the overall accessibility of the work, allowing readers to grasp the key ideas without difficulty.
- Extensive experiments to evaluate the effectiveness of the proposed method. The experiments are conducted on 8 classification benchmarks and 6 retrieval benchmarks.

**Weaknesses:**

- The proposed Semantic Concepts Mining method is not novel. Previous work [1] has already used CLIP as a concept labeler to construct pseudo labels for contrastive learning and image captioning. While [1] conducted experiments under the pretraining setting, this paper focuses on small domain datasets.
- Caption-level trapezoidal consistency builds incrementally on CyCLIP [2] in a semi-supervised setting. CyCLIP introduced cross-modal and in-modal consistency. While the paper does provide some comparisons between CyCLIP and SEMICLIP, it primarily emphasizes SEMICLIP focuses on unlabeled images and surrogate captions. However, this change may not be particularly novel.
- The experiments are primarily conducted on specific domain datasets. What about the performance on general benchmarks, such as the retrieval benchmarks Flicker30k and COCO, as well as classification on ImageNet?

[1] Huang, Runhui, et al. "Nlip: Noise-robust language-image pre-training." Proceedings of the AAAI Conference on Artificial Intelligence. Vol. 37. No. 1. 2023.

[2] Goel, Shashank, et al. "Cyclip: Cyclic contrastive language-image pretraining." Advances in Neural Information Processing Systems 35 (2022): 6704-6719.

**Questions:**

Please refer to the weaknesses section to see the exact questions.

---

> ### Author Response · Authors · 2024-11-16
>
> Dear Reviewer WcrK,
>
> We appreciate the reviewer for the thoughtful reviews and the encouraging comments such as *clear manner*, *easy to understand*, and *grasp the key ideas without difficulty*. Following are our responses to the concerns.
>
> **Weakness #1: Semantic Concepts Mining method is not novel.**
>
> **Response:**  During the semantic concepts mining (SCM), the model is **adapted to a task-specific domain** and trained with a classifier that **aligns with the concepts in the images**, which lays an important foundation for generating surrogate captions in the second stage. We believe SCM is an **innovative approach to utilizing unlabeled data**, and significant improvement in model performance is achieved through image-concept alignment in Figure (2a) and Figure (2b).
>
> Compared to concept labeler in [1], the differences are as follows:
>
> (1) We train a linear classifier to achieve better image-concept alignment in task-specific domain, while concept labeler in [1] obtains concepts through image-concept retrieval, which may not enable effective retrieval in certain specialized domains due to the niche nature of some concepts.
>
> (2) Concepts generated in [1] are considered as conditional input to the cross-modal decoder, which requires a large amount of data for training, significantly increasing the overhead. However, SemiCLIP can be trained with only a small amount of labeled data, achieving significant improvements in model performance in semi-supervised settings.
>
> Given the similarities between [1] and our method, we will cite [1] in the revised paper and include it in the related work section.
>
> [1] Nlip: Noise-robust language-image pre-training
>
> **Weakness #2: Caption-level trapezoidal consistency builds incrementally on CyCLIP.**
>
> **Response:** While there is a formal similarity between SemiCLIP and CyCLIP, we argue that the real innovation is found in the **deeper insights and practical applications** of the method, rather than its form. Therefore, we summarize the innovation and contribution of caption-level trapezoidal consistency (CTC) below:
>
> (1) CTC introduces CyCLIP into the semi-supervised learning scenario by utilizing unlabeled images and surrogate captions, thereby **extending the applicability** of CyCLIP beyond complete image-text pairs.
>
> (2) We found that CyCLIP can partially **alleviate the impact of noise** in the captions on training and provided a **geometric explanation**. From experiments in Figure (2c), if we directly constrain the reduction of the lower base of trapezoid, the model's performance will decrease by an average of 4.07%. This indicates that the direct alignment of unlabeled images and surrogate captions suffers from performance degradation due to the noise present in surrogate captions, whereas trapezoidal consistency effectively mitigates this issue and significantly improves performance through interactions between samples from both image and text modalities.
>
> (3) CTC offers new insights for future work **addressing less-than-ideal alignment** in image-text pairs, which is a widespread issue in practical settings.

---

> > ### Author Response · Authors · 2024-11-16
> >
> > **Weakness #3: The performance on general benchmarks .**
> >
> > **Response:** Our method aims to adapt the model to a  task-specific domain by leveraging a small amount of labeled data and a large number of unlabeled images and realize a notable enhancement in model performance within this domain. However, it is evident that after the adaptation, the model may **lose some of its broader generalization capability** [1,2] due to catastrophic forgetting, so the model's performance on common zero-shot classification and multimodal retrieval is **unlikely to achieve promising results**.
> >
> > However, we believe that in semi-supervised settings, adapting to a task-specific domain while maintaining the pre-trained model's original generalization ability will be an intriguing research topic, and we may explore this area more deeply in the future.
> >
> > In addition, to evaluate the performance of our method on general benchmark, we conducted experiments on CoCo, and the averaged retrieval results are as follows:
> >
> > |       CoCo        | I2T  | T2I  |
> > | :---------------: | :--: | :--: |
> > | CLIP (fine-tuned) | 50.3 | 50.9 |
> > |      S-CLIP       | 46.9 | 45.4 |
> > |     SemiCLIP      | 55.9 | 56.2 |
> >
> > CLIP (fine-tuned) refers to CLIP fine-tuned using only labeled data. I2T and T2I represent image→text retrieval and text→image retrieval respectively. The results indicate that SemiCLIP can achieve significant performance improvements on general benchmark over CLIP (fine-tuned) and  S-CLIP.  It is worth noting that S-CLIP's performance shows an average decrease of 4.5% compared to CLIP (fine-tuned), aligning with the paper's claim [3] that S-CLIP experiences performance drops when trained on a small number of image-text pairs in common datasets like CoCo. However, the superior performance of our proposed SemiCLIP is unaffected by the different types of datasets, achieving significant improvements on both commonly used datasets and task-specific datasets.
> >
> > For datasets like ImageNet, which is a classification dataset, it is not well-suited for image-text contrastive learning and thus does not align with the scenarios addressed in this paper.
> >
> > [1] Fine-Tuning can Distort Pretrained Features and Underperform Out-of-Distribution
> >
> > [2] An empirical study of catastrophic forgetting in large language models during continual fine-tuning
> >
> > [3] S-CLIP: Semi-supervised Vision-Language Learning using Few Specialist Captions

---

> ### Author Response · Authors · 2024-11-25
>
> Dear Reviewer WcrK,
>
> We sincerely thank the reviewer for valuable comments. We have addressed them in our responses and updated the manuscript accordingly. If the reviewer has any further questions, we are always ready to provide additional clarifications.
>
> Best regards,
>
> The Authors

---

> > ### Comment · Reviewer_WcrK · 2024-12-01
> >
> > Thanks to the authors for the rebuttal. However, I still have concerns regarding W2 and W3.
> >
> > For W2, in my opinion, extending CyCLIP to images and pseudo captions seems trivial.
> >
> > For W3, the table does not include CLIP's performance on COCO zero-shot retrieval. However, according to the CLIP paper, I found that the R1 scores for image retrieval and text retrieval are 58.4 and 37.8, respectively. The CLIP fine-tuning performance provided in the rebuttal appears to degrade performance on image retrieval while improving performance on text retrieval, which seems counterintuitive. Additionally, SemiCLIP also reduces performance on image retrieval, suggesting that using semi-supervision on a general dataset harms the model’s original generative capability, which makes me question the effectiveness of the proposed method. However, according to [1], fine-tuning on the COCO training set can improve performance from 58.6 to 77.0 on image retrieval and from 45.6 to 59.9 on text retrieval.
> >
> > Regarding to the rebuttal, I towards maintaining my score.
> >
> > [1] Scaling Up Visual and Vision-Language Representation Learning With Noisy Text Supervision

---

> > > ### Author Response · Authors · 2024-12-01
> > >
> > > Dear Reviewer WcrK,
> > >
> > > We sincerely thank you for your effort in reviewing our paper and would like to provide additional clarification regarding the concerns you mentioned.
> > >
> > > **Concern #1: Extending CyCLIP to images and pseudo captions seems trivial.**
> > >
> > > **Response:** First of all, the main contribution of our work is presenting a new framework for adapting CLIP model to specific domain tasks using semi-supervision. As a part of the proposed framework, the caption-level trapezoidal consistency module extends CyCLIP to images without ground-truth captions by new techniques. Additional, our method mitigates noise in pseudo captions, offering insights into addressing alignment issues in image-text pairs. We believe it is meaningful to extend known techniques to new tasks in simple ways, which can lead to significant performance enhancements.
> > >
> > > Therefore, although we agree with the reviewer that our trapezoidal consistency loss builds upon CyCLIP, **the learning framework which significantly improves CLIP alignment capability in specific domain tasks using semi-supervision is not trivial**.
> > >
> > >
> > > **Concern #2: The results related to COCO dataset.**
> > >
> > > **Response:** Thank you for your insightful and detailed observation. We would like to address the reviewer's concern from three perspectives:
> > >
> > > (1) The results reported in [1] are likely based on the **ViT-L/14 CLIP model**, which is significantly more powerful than the **ViT-B/16 CLIP model** we used in our experiments. Unfortunately, due to time constraints, we are unable to provide results for ViT-L/14 CLIP model.
> > >
> > > (2) We provide the retrieval performance for the zero-shot CLIP (ViT-B/16) on COCO dataset in the table below:
> > >
> > > |  COCO (1% labeled data)  | I2T R1 | I2T R5 | T2I R1 | T2I R5 |
> > > | :---------------: | :----: | :----: | :----: | :----: |
> > > | CLIP (zero-shot)  |  35.5  |  60.7  |  33.1  |  57.3  |
> > > | CLIP (fine-tuned using only labeled data) |  33.7  |  60.2  |  33.6  |  60.5  |
> > > |      S-CLIP       |  29.5  |  54.8  |  27.8  |  53.8  |
> > > |     SemiCLIP      |  **37.8**  |  **64.4**  |  **37.6**  |  **65.7**  |
> > >
> > > From the results, we can see that SemiCLIP outperforms zero-shot CLIP in retrieval performance by an average of 4.7%, indicating that **the use of semi-supervision does not harm the model's original generalization ability on the general dataset**. By contrast, our main competitor, S-CLIP, shows a performance decline of 5.2% compared to zero-shot CLIP, further highlighting the superiority of our approach.
> > >
> > > (3) As stated in Appendix E.1 of S-CLIP [2], for general datasets, "fine-tuning models using limited image-caption pairs degrades performance, as the original CLIP already performs well", we find that the performance gains obtained by SemiCLIP on general datasets are indeed less pronounced compared to specific domain datasets. We believe this observation can inspire future studies on the CLIP model adapting to achieve a better performance trade-off between general and specific domain datasets.
> > >
> > > We thank the reviewer again for the insightful comments, and if there are any other questions, we are always ready to provide additional clarifications.
> > >
> > >
> > > [1] Scaling Up Visual and Vision-Language Representation Learning With Noisy Text Supervision
> > >
> > > [2] S-CLIP: Semi-supervised Vision-Language Learning using Few Specialist Captions

---

> > > > ### Comment · Reviewer_WcrK · 2024-12-01
> > > >
> > > > Regarding concern #1, the concern still remains.
> > > > Regarding concern #2, since the model is ViT-B/16 and the results and settings are quite different from those in the S-CLIP paper (which also conducts experiments on COCO), I could not assess the reliability of the experiments.

---

> > > > > ### Author Response · Authors · 2024-12-01
> > > > >
> > > > > Thank you for your response. We provide additional clarification for your concern #2 below:
> > > > >
> > > > > (1) S-CLIP has acknowledged its limitations in achieving satisfactory performance on general datasets in the appendix. The COCO results in S-CLIP are obtained by training on a subset of COCO, specifically the "sports" category, chosen for a specialized domain task. As the details of this subset selection are not publicly available, we are unable to replicate S-CLIP's experiments. In addition, experiments conducted only on a subset of COCO cannot validate the method's performance on general datasets. In our COCO experiments, we utilized the complete COCO dataset and selected 1% data as labeled data, which accounts for the differences in numerical results. We confidently affirm the reliability of our experimental results.
> > > > >
> > > > > (2) We provide the code to reproduce the COCO dataset results in the anonymous github (https://anonymous.4open.science/r/SemiCLIP_COCO-0336).

---

### Official Review · Reviewer_zoSv · 2024-11-04

**Soundness:** 3
**Presentation:** 3
**Contribution:** 3
**Rating:** 6
**Confidence:** 4

**Summary:**

-- This paper proposes a new semi-supervised CLIP training method SEMICLIP that adapts CLIP to target tasks using only a small amount of image-text pairs.

-- This paper designs concept-level consistency and caption-level trapezoidal consistency for learning from unlabeled images to enhance visual representations and improve cross-modal alignment, respectively.

-- Extensive experiments demonstrate that the proposed method achieves state-of-the-art results in both zero-shot classification and image-text retrieval tasks.

**Strengths:**

-- SEMICLIP mines candidate semantic concepts from labeled data and learns to associate images with concepts, which may be a new way of using unlabeled images in vision-language pre-training.

-- This paper proposes the trapezoidal consistency to enhance the multi-modal alignment by exploiting the geometric structure of trapezoids in the representation space.

-- This paper uses prompts-driven templates and predicts concepts to construct surrogate captions for unlabeled images.

**Weaknesses:**

-- The proposed method consists of two stages. The finetune stage heavily relies on the concept ming from the pretraining stage. The contribution of each stage is hard to evaluate.

-- The experiments are only conducted on task-specific datasets. What is their performance on common zero-shot classification and multimodal retrieval?

**Questions:**

-- In SEMANTIC CONCEPTS MINING,  the linear classifier is initialized from the concept features of clip text encoder. Do the parameters of this module update during pre-training?

-- It seems this linear classifier is a close-set classifier, why not use the open-set method?

-- Does each concept have its own prompts-driven template [V ]1 [V ]2 [V ]3? Is the prompts-driven template [V ]1 [V ]2 [V ]3 shared for all concepts?

-- It is unclear whether the model is trained from scratch or initialized from pre-trained CLIP.

-- The sizes of datasets in the experiments are unclear.

-- Why do the proposed models suppress the fine-tuned models (CLIP (fine-tuned))? And what is the upper bound for performance?

-- Why is the lower base not used in Figure 1(b)?

---

> ### Author Response · Authors · 2024-11-16
>
> Dear Reviewer zoSv,
>
> We sincerely appreciate the reviewer for thoughtful feedback. We are encouraged for comments like *a new way of using unlabeled images*. We address your concerns one by one.
>
> **Weakness #1: The contribution of each stage.**
>
> **Response:** In the first stage, the model is trained using the contrastive loss from CLIP with labeled data, along with the soft cross-entropy loss for the semantic linear classifier as defined in Eq. (2). During this stage, the model is **adapted to a task-specific domain** and trained with a classifier that **aligns with the concepts in the images**, which lays an important foundation for generating surrogate captions in the second stage. It is worth noting that in Figures (2a) and (2b) of the paper, we observed a **significant improvement** in model performance after the first stage (i.e., the SPT stage) compared to fine-tuning CLIP, indicating that the alignment of concepts with the images during this phase contributes to enhanced visual representation capabilities. In the second stage, SemiCLIP achieves **further performance improvement** by leveraging concept-level semantic consistency and caption-level trapezoidal consistency.
>
> The performance variations across the two stages and comparisons with other methods are presented below:
>
> |     Zero-shot     | Remote Sensing | Fashion | RS(L$\neq$U) |
> | :---------------: | :------------: | :-----: | :----------: |
> | CLIP (fine-tuned) |      79.7      |  46.8   |     80.8     |
> |      S-CLIP       |      84.0      |  54.2   |     82.1     |
> |      Stage1       |      81.3      |  55.1   |     82.3     |
> | Stage2 (SemiCLIP) |      85.7      |  60.8   |     85.0     |
>
> |     Retrieval     | Remote Sensing | Fashion | SciCap | Simpsons |
> | :---------------: | :------------: | :-----: | :----: | :------: |
> | CLIP (fine-tuned) |      28.8      |  14.3   |  14.8  |   31.9   |
> |      S-CLIP       |      29.6      |  19.1   |  16.5  |   31.0   |
> |      Stage1       |      29.3      |  20.7   |  15.1  |   33.2   |
> | Stage2 (SemiCLIP) |      31.7      |  24.3   |  17.0  |   37.8   |
>
> CLIP (fine-tuned) refers to CLIP fine-tuned using only labeled data. From the table above, Stage 2 achieved an average performance increase of 3.6% over Stage 1. In addition, Stage 1 outperformed CLIP (fine-tuned) by an average of 2.8%, indicating that the alignment of concepts with images helps improve visual representation.
>
> **Weakness #2: The performance on common zero-shot classification and multimodal retrieval.**
>
> **Response:** Our method aims to adapt the model to a task-specific domain by leveraging a small amount of labeled data and a large number of unlabeled images and realize a notable enhancement in model performance within this domain. However, it is evident that after the adaptation, the model may **lose some of its broader generalization capability** [1,2] due to catastrophic forgetting, so the model's performance on common zero-shot classification and multimodal retrieval is **unlikely to achieve promising results**.
>
> However, we believe that in semi-supervised settings, adapting to a task-specific domain while maintaining the pre-trained model's original generalization ability will be an intriguing research topic, and we may explore this area more deeply in the future.
>
> In addition, to evaluate the performance of our method on common dataset, we conducted experiments on CoCo, and the averaged retrieval results are as follows:
>
> |       CoCo        | I2T  | T2I  |
> | :---------------: | :--: | :--: |
> | CLIP (fine-tuned) | 50.3 | 50.9 |
> |      S-CLIP       | 46.9 | 45.4 |
> |     SemiCLIP      | 55.9 | 56.2 |
>
> I2T and T2I represent image→text retrieval and text→image retrieval respectively. The results indicate that SemiCLIP can achieve significant performance improvements on common datasets over CLIP (fine-tuned) and  S-CLIP.  It is worth noting that S-CLIP's performance shows an average decrease of 4.5% compared to CLIP (fine-tuned), aligning with the paper's claim [3] that S-CLIP experiences performance drops when trained on a small number of image-text pairs in common datasets like CoCo. However, the superior performance of our proposed SemiCLIP is unaffected by the different types of datasets, achieving significant improvements on both commonly used datasets and task-specific datasets.
>
> [1] Fine-Tuning can Distort Pretrained Features and Underperform Out-of-Distribution
>
> [2] An empirical study of catastrophic forgetting in large language models during continual fine-tuning
>
> [3] S-CLIP: Semi-supervised Vision-Language Learning using Few Specialist Captions

---

> > ### Author Response · Authors · 2024-11-16
> >
> > **Question #1: The update of linear classifier for Semantic Concepts Mining.**
> >
> > **Response:** The parameters of linear classifier are updated during training. The initialization of the CLIP text encoder provides a rich semantic foundation for the linear classifier. With further fine-tuning, the linear classification head can achieve better concept mining within the task-specific domain. The training loss function for linear classifer is Eq. (2) and we will elaborate on this point more clearly in the main paper.
> >
> > **Question #2: Why not use the open-set methods for linear classifier training.**
> >
> > **Response:** We answer this question by developing an open-set classifier. Specifically, through the first stage of CLIP loss training with labeled data, we leverage the model's zero-shot capabilities to assign corresponding concepts to each image, which performs the role of an open-set classifier. The concepts list is still extracted from the captions of the labeled data, and for each image, we select the concepts with top-k cosine similarity. The second stage is consistent with SemiCLIP, except that the concepts are generated by open-set classifier. The experimental results of averaged performance on remote sensing and fashion datatsets are as follows:
> >
> > |         Remote sensing          |  ZS  | I2T  | T2I  |
> > | :-----------------------------: | :--: | :--: | :--: |
> > | SemiCLIP (Close-set classifier) | 85.7 | 32.4 | 31.1 |
> > | SemiCLIP (Open-set classifier)  | 82.6 | 32.0 | 30.8 |
> >
> > |             Fashion             |  ZS  | I2T  | T2I  |
> > | :-----------------------------: | :--: | :--: | :--: |
> > | SemiCLIP (Close-set classifier) | 60.8 | 24.2 | 24.5 |
> > | SemiCLIP (Open-set classifier)  | 56.9 | 23.5 | 23.6 |
> >
> > In the table, we report the average performance of ZS (zero-shot), I2T (image→text retrieval), and T2I (text→image retrieval) on the remote sensing and fashion datasets. The results reveal a performance decline when using the open-set classifier. We attribute this to the fact that some concepts in the task-specific domain are not well captured by the image-text alignment model, making it difficult to effectively generate the corresponding concepts. We noticed a significant decline in performance of zero-shot, with a **drop of 3.5%**. This suggests that close-set classifier is more robust to these task-specific concepts due to its image-concept alignment, resulting in better performance.
> >
> > **Question #3: The sharing for prompts-driven template.**
> >
> > **Response:** The prompts-driven templates are shared among all concepts, which is designed to help surrogate captions better adapt to the specific tasks. CoOp [1] also leverages shared prompts to assist the model's adaptation. While shared prompts are generally sufficient for most cases, crafting prompts specific to different concepts could yield better results in more complex scenarios, which should be explored further in future studies.
> >
> > [1] Learning to Prompt for Vision-Language Models
> >
> > **Question #4: It is unclear whether the model is trained from scratch or initialized from pre-trained CLIP.**
> >
> > **Response:** The model is initialized from pre-trained CLIP, which allows the model to leverage CLIP's rich semantics and strong generalization capabilities, enabling it to quickly adapt to a variety of downstream tasks. In practice, considering that semi-supervised learning typically involves a small amount of labeled data, training from scratch is often not very feasible for large models.
> >
> > Using the pre-trained CLIP initialization weights follows the previous approach S-CLIP [1] and is also a common practice in adapting pre-trained models for downstream tasks. All comparison methods in the paper use a pre-trained CLIP model for initialization, ensuring that the experiments are fair.
> >
> > [1] S-CLIP: Semi-supervised Vision-Language Learning using Few Specialist Captions

---

> ### Author Response · Authors · 2024-11-16
>
> **Question #5: The sizes of datasets in the experiments.**
>
> **Response:** For remote sensing datasets, the RSICD, UCM, and Sydney datasets contain 8734, 1680, and 497 image-text pairs, respectively. The three datasets constitute RS-ALL, and we randomly select 10% of the image-text pairs from the training set as labeled data, with the remaining pairs treated as unlabeled. During the inference, the sizes for classification datasets are below:
>
> |                   | RSICD-CLS | UCM-CLS | WHU-RS19 | RSSCN7 |  AID  |
> | :---------------: | :-------: | :-----: | :------: | :----: | :---: |
> | Number of images  |   1094    |  2100   |   1005   |  2800  | 10000 |
> | Number of classes |    31     |   21    |    19    |   7    |  30   |
>
> For fashion datasets, the Fashion200k, FashionGen, and Polyvore datasets contain 61753, 60147, and 71967 image-text pairs, respectively. The sizes for classification datasets are below:
>
> |                   |         Fashion200k         |          FashionGen           | Polyvore |
> | :---------------: | :-------------------------: | :---------------------------: | :------: |
> | Number of images  |            29785            |             32528             |  14657   |
> | Number of classes | Super-class 5, Sub-class 31 | Super-class 48, Sub-class 121 |    11    |
>
> For other datasets, SciCap contains 106934 image-text pairs and Simpsons only contains 720 pairs.
>
> We will provide further details on the sizes of the datasets in the paper.
>
> **Question #6: Why do the proposed model suppress CLIP (fine-tuned) and the upper bound for performance.**
>
> **Response:** The poor performance of CLIP (fine-tuned) can be attributed to the fact that it was trained solely on labeled data. The possible upper bound of the performance is supervised learning on all training data, which we refer it as Oracle (fully supervised fine-tuned). We show the averaged oracle performance below:
>
> |            Remote sensing            |  ZS  | I2T  | T2I  |
> | :----------------------------------: | :--: | :--: | :--: |
> |               SemiCLIP               | 85.7 | 32.4 | 31.1 |
> | Oracle (fully supervised fine-tuned) | 82.0 | 39.9 | 36.6 |
>
> |               Fashion                |  ZS  | I2T  | T2I  |
> | :----------------------------------: | :--: | :--: | :--: |
> |               SemiCLIP               | 60.8 | 24.2 | 24.5 |
> | Oracle (fully supervised fine-tuned) | 58.5 | 37.4 | 37.0 |
>
> Interestingly, our method even outperforms the oracle calculated here in zero-shot performance. We believe this is due to the severe imbalance in the proportions of the sub-datasets within the training sets, RS-ALL and Fashion. When training on the full dataset, sub-datasets with a smaller proportion, such as UCM, perform poorly on their corresponding zero-shot test set, UCM-CLS, which resulted in a slightly lower overall performance. We find this to be an interesting phenomenon, and it warrants further research into how to mitigate the issue of imbalance in the proportions of the sub-datasets.
>
> The oracle outperforms SemiCLIP in retrieval performance, indicating that increasing the amount of labeled data will significantly improve retrieval performance. This also points to a potential direction for further improvement of SemiCLIP in the future. We will include the oracle results in the experimental tables of the paper, helping readers gain a clearer understanding of the task.
>
> **Question #7: Why is the lower base not used in Figure 1(b)?**
>
> **Response:** In caption-level trapezoidal consistency, since surrogate captions for unlabeled images may not be reliable, we avoid directly constraining their relationship and do not directly use the lower base. In fact, by imposing constraints on the upper base, legs, and diagonals, we enable the interaction between in-modal and cross-modal samples, ensuring coherence among samples within each modality and substantially enhancing the model's overall alignment ability.
>
> In Figure (2c), if we directly constrain the reduction of the lower base, the model's performance will decrease by an average of 4.07%. This indicates that the direct alignment of unlabeled images and surrogate captions suffers from performance degradation due to the noise present in surrogate captions, whereas trapezoidal consistency effectively mitigates this issue and significantly improves performance through interactions between samples from both image and text modalities.

---

> > ### Comment · Reviewer_zoSv · 2024-11-20
> >
> > Thanks for your responses. I have read the rebuttal and think an overall rating 6 is reasonable.

---

> > > ### Author Response · Authors · 2024-11-20
> > >
> > > Thank you for taking the time to review our responses and for your thoughtful evaluation. We appreciate your efforts in assessing our work.

---

### Author Response · Authors · 2024-11-18
**Summary of Manuscript Changes and Looking Forward to Discussing with Reviewers**

We sincerely appreciate all reviewers for their time and effort. The valuable comments have been instrumental in improving our paper. In addition to our responses, we have also made updates to the manuscript to address concerns. The details are as follows:

(1) For avoid misunderstanding, we provide a clearer explanation of the training of the linear classifier $\psi$ in Section 3.2.

(2) To illustrate the upper bound of performance, we add the results of fully supervised training to Tables 1-5 and provide the corresponding analysis in Appendix E.

(3) In order to provide a clear view of the data, we include a description of the dataset size in Appendix F.

(4) To compare with the differences from Nlip, we cite and compare it in Appendix G.

(5) In order to make the title more aligned with the paper’s content, we replaced 'training' with 'adaptation'.

(6) To highlight the role of the prompting strategy, we compare the effects of different prompt positions, initializations, and learnability on performance in Appendix H.

(7) To evaluate the robustness of SemiCLIP, we conduct ablation study regarding different percentages of labeled data in Appendix I.

(8) To evaluate the performance of SemiCLIP on general benchmarks, we provoide results for COCO datasets in Appendix J.

(9) To emphasize the novelty of trapezoidal consistency, we further elaborate in Appendix C on its differences from CyCLIP and the reasons for its effectiveness.

We are open to any additional questions or feedback reviewers may have.

Best Regards,

Authors

---

### Meta-Review · Area_Chair_fbTS · 2024-12-20

**Metareview:**

(a) Scientific Claims and Findings

The paper introduces SEMICLIP, a semi-supervised training method for CLIP models that enhances performance using limited image-text pairs. SEMICLIP employs semantic concept mining to create pseudo-labels for unlabeled data and introduces trapezoidal consistency regularization to maintain geometric relationships between image-text pairs. The method is evaluated on zero-shot classification and image-text retrieval tasks, showing improvements over existing baselines. Reviewers highlight the method's potential to improve cross-modal alignment and visual representation using unlabeled images.

(b) Strengths

Reviewer zoSv appreciates the novel approach of using semantic concepts and trapezoidal consistency to enhance visual representations and cross-modal alignment. WcrK commends the clear writing and extensive experiments across multiple benchmarks. 52fH notes the innovative use of trapezoidal consistency loss, while Psom highlights the intuitive correlation between images and concepts. JQns finds the paper well-written, with comprehensive experiments showing consistent improvements over baselines.

(c) Weaknesses

The reviewers identify several weaknesses. zoSv points out the difficulty in evaluating the contribution of each training stage and the limited scope of datasets used. WcrK questions the novelty of the semantic concepts mining method and the incremental nature of trapezoidal consistency. 52fH suggests that the method for mining semantic concepts is not novel and questions the rationale behind trapezoidal consistency. Psom raises concerns about the prompting strategy and the quality of pseudo-labels. JQns finds the motivation for trapezoidal consistency insufficiently explained and questions the surrogate caption generation process.

(d) Decision Reasons

On balance, AC agrees with positive points raised by the reviewers which outweigh the negative ones. The decision to accept the paper is based on its approach to semi-supervised training for CLIP models and the promising experimental results. The method's ability to leverage unlabeled images and improve cross-modal alignment is a significant contribution, as highlighted by reviewers zoSv and JQns. While there are concerns about the novelty of certain components and the explanation of trapezoidal consistency, the overall strengths in innovation, experimental validation, and potential impact outweigh these weaknesses. The paper's contributions to enhancing CLIP's adaptability and performance make it a valuable addition to the conference.

**Additional Comments On Reviewer Discussion:**

During the rebuttal period, the authors addressed several concerns raised by the reviewers, leading to some adjustments in their evaluations.

Reviewer zoSv expressed satisfaction with the authors' responses and considered a "weak accept" rating to be reasonable, indicating that the rebuttal addressed their concerns adequately.

Reviewer WcrK maintained concerns regarding the novelty of extending CyCLIP to images and pseudo captions, as well as the performance on the COCO dataset. They noted that the results seemed counterintuitive and questioned the effectiveness of the proposed method. Despite the authors' rebuttal, WcrK decided to maintain a "weak reject" rating due to unresolved concerns about the reliability of the experiments.

Reviewer 52fH found the authors' explanations technically sound and decided to raise their rating, indicating that the rebuttal successfully addressed their concerns.

Reviewer Psom stated that the response addressed their concerns and leaned towards accepting the paper, showing a positive reception to the authors' efforts.

Reviewer JQns appreciated the additional experiments and clarifications provided by the authors, which resolved their concerns. They updated their rating accordingly and recommended acceptance, noting that the authors effectively addressed the main concerns regarding motivation, design choices, and performance on general benchmarks. JQns acknowledged that while the novelty of the proposed method was questioned, the adaptation of existing methods to new contexts constituted a meaningful contribution.

In weighing these points for the final decision, the authors' ability to address most reviewer concerns effectively during the rebuttal period was a significant factor. The positive feedback from reviewers zoSv, 52fH, Psom, and JQns, who acknowledged that their concerns were resolved, reinforced the decision to accept the paper. Despite WcrK's remaining concerns, the overall consensus and the meaningful contributions highlighted by JQns supported the paper's acceptance.

---

### Decision · Program_Chairs · 2025-01-22

Accept (Poster)